# The RNA-binding protein Squid regulates embryonic midgut development via *Axin* alternative splicing in *Bombyx mori*
Chunmei Tong[1,2,4], Wanyu Mo[2,4], Minling Cai[2], Yuling Peng[2], Jilei Huang[3], Kang Li [2]✉ & Huimin Deng [2]✉

The insect midgut is crucial for digestion and nutrient absorption, but its embryonic development is poorly understood. Here, we show that the RNA-binding protein Squid is essential for embryonic midgut epithelium development in the model insect, *Bombyx mori*. CRISPR/Cas9-mediated knockout of *Squid* causes embryonic lethality, with mutants exhibiting disorganized midgut epithelium, lipid droplet accumulation, and impaired nutrient absorption. Integrated RNA sequencing (RNA-seq) and RNA-immunoprecipitation sequencing (RIP-seq) analyses reveal that Squid directly regulates the alternative splicing of *Axin*, a key Wnt/β-catenin pathway component. In *Squid*-depleted embryos, *Axin* splicing shifts from the long (*Axin-L*) to a short isoform (*Axin-S*), as confirmed by RT-qPCR. Consequently, β-catenin protein levels are significantly reduced in the midgut epithelium. Overexpression in *Bm*N cells confirms that Axin-L, but not Axin-S, elevates β-catenin. Overall, this study uncovers a critical post-transcriptional pathway wherein Squid ensures proper midgut development by regulating *Axin* alternative splicing to fine-tune β-catenin levels.

Insects represent the dominant group of terrestrial animals in both species diversity and abundance. Their evolutionary success is demonstrated by their colonization of numerous ecological niches and utilization of diverse food sources. These varied feeding strategies are reflected in corresponding morphofunctional adaptations of their alimentary canal[1].

Despite morphological variations, the insect alimentary canal is characterized by a single layer of epithelial cells supported by a basal lamina encircled by muscular layers. It is segmented into three principal regions: the foregut, midgut and hindgut, each with distinct characteristics, functions, and embryonic origins[2,3]. Insect midgut is pivotal for digestion and nutrient absorption, and it also serves critical roles in immune response, cellular regeneration, and recovery from luminal infections[2–4]. Consequently, the insect midgut has emerged as a valuable model for studying tissue recovery following pathogen-induced injury[5,6] and tissue remodeling during metamorphosis[7–9]. Moreover, it represents a prime target for pest control strategies, particularly with the advent of RNA interference (RNAi) technology[10,11]. However, the molecular and cellular mechanisms underlying embryonic midgut development in insects remain largely elusive.

Heterogeneous nuclear ribonucleoproteins (hnRNPs) are a large family of RNA-binding proteins with essential functions in alternative splicing, mRNA stabilization, transcription, and translation[12]. A key member of this family, hrp40, also known as Squid, is crucial for RNA metabolism across species. In mammals, the Squid homolog hnRNPD regulates the nucleocytoplasmic shuttling of Alpha-synuclein (SNCA) transcripts, implicated in the pathogenesis of Parkinson's disease[13] and participates in macrophage polarization via the lncRNA MCP1 axis[14]. In insects, *Squid* mutation disrupts dorsal-ventral patterning by mislocalizing *grk* mRNA during oogenesis and is essential for establishing oocyte polarity through *osk* mRNA localization[15–18]. Furthermore, Squid plays a crucial role in the regulation of pre-mRNA alternative splicing. For instance, Squid interacts with hrp38 and pADPr to inhibit the alternative splicing of *Hsrx-RC* pre-mRNA, while it enhances the alternative splicing of *Ddc* pre-mRNA[19]. Squid cooperates with other hnRNPs to regulate the alternative splicing of multiple mRNA and is contributed to sex-specific splicing regulation in *Drosophila*[20,21]. Our previous study demonstrates that Squid regulates the alternative splicing of the *POUM2* gene, which is essential for embryonic cuticular formation and tanning in the model insect, *Bombyx*

[1]College of Life Sciences, Zhaoqing University, Zhaoqing, China. [2]Guangdong Key Laboratory of Insect Developmental Biology and Applied Technology, Guangzhou Key Laboratory of Insect Development Regulation and Application Research, Institute of Insect Science and Technology & School of Life Sciences, South China Normal University, Guangzhou, China. [3]Instrumental Analysis and Research Center, South China Agricultural University, Guangzhou, China. [4]These authors contributed equally: Chunmei Tong, Wanyu Mo. ✉e-mail: likang@m.scnu.edu.cn; denghuiminmin@163.com

mori[22]. However, the physiological functions of Squid-mediated pre-mRNA alternative splicing remain to be further revealed.

In this study, we elucidate the physiological role of the RNA-binding protein Squid in pre-mRNA alternative splicing during *B. mori* embryonic development. Integrated RNA-seq and RIP-seq analyses in a *Squid* mutant reveal that Squid directly regulates the alternative splicing of *Axin*, a core component of the Wnt/β-catenin pathway. This regulation is necessary for maintaining normal β-catenin expression levels in the basal midgut epithelium. Our findings provide the critical evidence establishing an hnRNP family member as essential for insect embryonic midgut development.

## Results

### Squid is essential for embryonic development in *B. mori*

To elucidate the function of Squid in *B. mori*, we first characterized its domain architecture. *B. mori* Squid contains two RNA recognition motifs (RRMs) and a low-complexity region (Fig. 1A). Structural predictions indicated that each RRM adopts a typical fold consisting of two α-helices and four β-sheets (Fig. 1B). Sequence alignment further revealed that both insect and mammalian Squid orthologs possess two highly conserved RRMs (Supplementary Fig. 1), suggesting an evolutionarily conserved role in RNA binding.

Given our previous observation of high *Squid* mRNA expression during embryogenesis[22], we investigated its functional role in *B. mori* embryonic development. A *Squid* mutant line was generated using CRISPR/Cas9-mediated genome editing. Three 23 bp sgRNAs (Sites 1–3) were designed to target regions flanking the translation start site and synthesized in vitro (Supplementary Fig. 2A). A mixture of sgRNAs and Cas9 protein was microinjected into preblastoderm embryos, yielding a heterozygous mutant (*Squid*[+/−]). This mutant carries a two-base pair deletion within the *Squid* gene open reading frame (ORF) (Supplementary Fig. 2B), predicted to cause four amino acid translation errors and introduce a premature stop codon (Supplementary Fig. 2C).

Among the offspring of two heterozygous parents, approximately 30% of individuals exhibited obviously delayed embryonic development at

7–8 days post-oviposition, ultimately dying within the eggshell by days 9–10. In contrast, offspring of wild-type parents showed only a 5% mortality rate (Fig. 1C–E). Genomic sequencing of five dead embryos confirmed that all were homozygous mutants (*Squid*[−/−]) (Supplementary Fig. 2D). Western blot analysis of wild-type embryos detected two specific bands (31.68 kDa and 33.44 kDa), corresponding to the predicted Squid-1 and Squid-2 isoforms (the latter containing an additional 48 bp segment). In contrast, no Squid protein was detected in *Squid*[−/−] embryos (Fig. 1F), confirming successful knockout and the high specificity of the antibody. These results indicate that the homologous mutation (*Squid*[−/−]) in *Squid* leads to embryonic lethality, confirming its essential role in embryonic development in *B. mori*.

### Knockout of *Squid* inhibits embryonic midgut development

To determine the role of Squid in embryonic development, we first examined its spatial expression pattern during organogenesis using a polyclonal anti-Squid antibody. Immunohistochemistry revealed Squid localization in developing midgut epithelial cells and epidermal cells (Fig. 2A-C), implying its involvement in the development of both tissues. While Squid function in the epidermis was preliminarily addressed in our previous study[22], the current work focuses on its role in midgut development. Hematoxylin and eosin (HE) staining of 7-day-old wild-type and *Squid*[−/−] embryos showed tightly and regularly organized midgut epithelial cells in wild-type (Fig. 2D and D'), whereas those in *Squid*[−/−] mutants were disorganized, with the midgut lumen containing numerous lipid droplet-like vacuoles (orange arrows, Fig. 2E and E'). The lepidopteran midgut epithelium typically comprises three distinct cell types: intestinal stem cells (ISCs), columnar cells (CCs), and goblet cells (GCs)[4]. Transmission electron microscopy (TEM) further demonstrated that the wild-type epithelium consisted mainly of CCs and GCs, with a small number of pear-shaped ISCs (red arrows) located basally. CCs exhibited a well-developed brush border of microvilli (purple arrow) and numerous spherites (green arrow) (Fig. 2D"). In contrast, *Squid*[−/−] embryos exhibited abnormal cellular morphology:

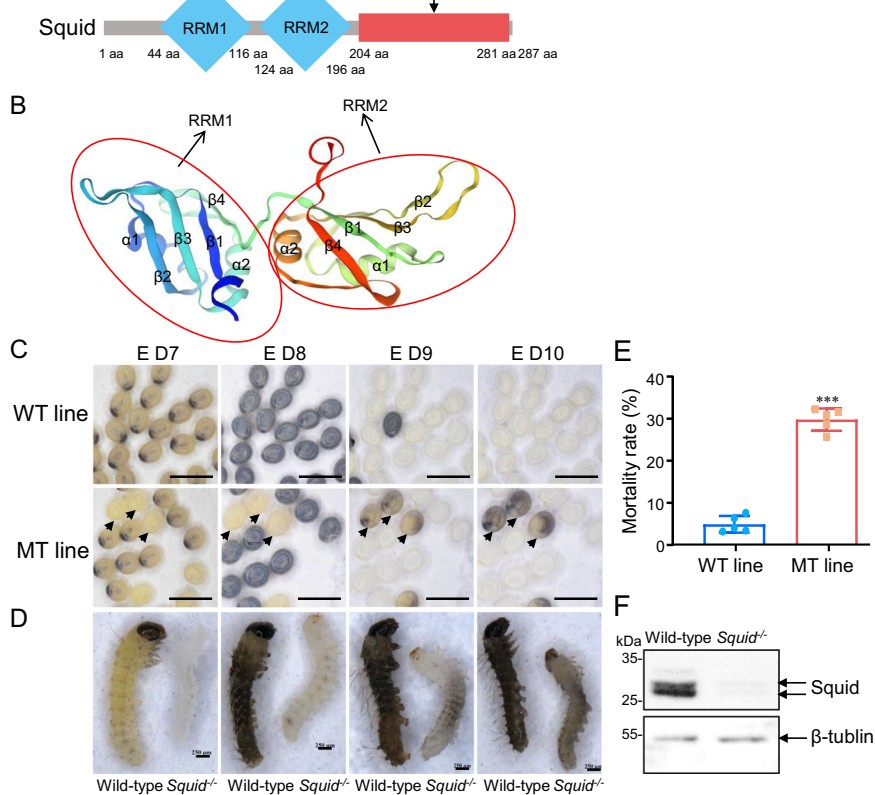

**Fig. 1 | CRISPR/Cas9-mediated knockout of *Squid* in silkworm. A** Schematic diagram of the domain architecture of Squid (Accession No.: D38013.1). aa, amino acid. **B** SWISS-MODEL diagram depicting the Squid structure, detailing the arrangement of α-helices and β-sheets. **C** Egg phenotypes from wild-type (WT) and heterozygous (MT) lines. Some of the offspring produced by two heterozygotes could not hatch, which is indicated by a black arrow. The WT line represents the offspring of two wild-type silkworms, and the MT line represents the offspring of two heterozygotes. The scale bars represent 5000 μm. **D** Representative images of dechorionated embryos from wild-type and *Squid*[−/−] mutants. The scale bars represent 250 μm. **E** Embryonic mortality rates in WT and MT lines. The data are presented as mean ± SD. n = 5 biologically independent samples. Statistical significance was determined by using *t* test. *p* < 0.05 (*), *p* < 0.01 (**) and *p* < 0.001 (***). **F** Western blot analysis of Squid expression in wild-type and *Squid*[−/−] embryos at day 7 of embryogenesis.

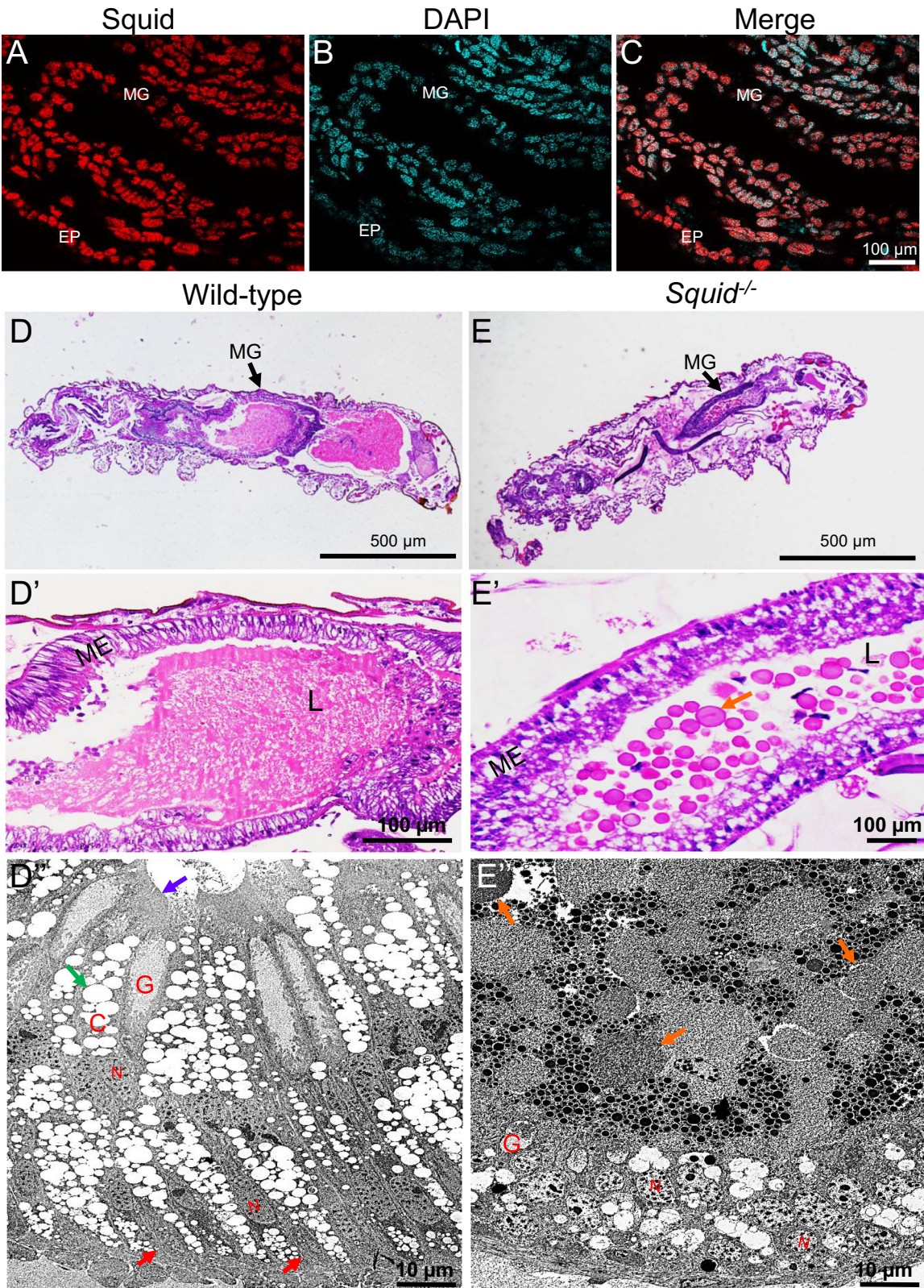

**Fig. 2 | Effect of *Squid* knockout on embryonic midgut development.**
**A–C** Localization of Squid in the embryonic midgut during organogenesis. MG, midgut. EP, epidermis. Red fluorescent signals indicate Squid. Nuclei are marked by DAPI in cyan. The scale bars represent 100 µm. **D–E'** Semi-thin cross-sections of the midgut in wild-type (**D–D'**) and *Squid*⁻/⁻ embryos (**E–E'**). The orange arrow represents lipid droplet-like vacuoles. MG, midgut. ME, midgut epithelium. L,

lumen. The scale bars represent 500 µm or 100 µm. **D''–E''** TEM images of the midgut epithelium in wild-type (**D''**) and *Squid*⁻/⁻ embryos (**E''**). C columnar cell, G goblet cell, N nucleus. The red and orange arrows represent intestinal stem cells (ISCs) and lipid droplets, respectively. The purple and green arrows represent microvilli and numerous spherites, respectively. The scale bars represent 10 µm.

only GCs with reduced cavities remained recognizable, while CCs and ISCs were poorly differentiated and could not be unambiguously identified (Fig. 2E"). These structural abnormalities indicate impaired midgut development in the absence of Squid. Additionally, abundant lipid droplets (orange arrow) were observed in the mutant midgut lumen (Fig. 2E"), implying that the nutrient absorption capacity of the midgut may be arrested following *Squid* knockout.

To determine whether *Squid* knockout affects ISC proliferation or differentiation, we performed HE staining and phospho-histone H3 (P-H3, a conserved cell proliferation marker) detection at earlier wild-type embryonic midgut developmental stages (days 5–day 6) and at day 7 in both wild-type and $Squid^{-/-}$ embryos. HE staining revealed that the wild-type midgut epithelium formed a single layer with an empty lumen on day 5 (Supplementary Fig. 3Aa), developed into a well-defined pseudostratified structure with numerous undigested lipid droplet-like vacuoles in the lumen by day 6 (Supplementary Fig. 3Ab), and exhibited a fully organized pseudostratified epithelium without obvious lipid droplet-like vacuoles by day 7 (Fig. 2D, D'). In contrast, the $Squid^{-/-}$ embryonic midgut epithelium appeared disorganized and contained abundant lipid-like vacuoles in the lumen at day 7 (Fig. 2E, E'). Moreover, P-H3 detection showed strong nuclear signals in nearly all midgut epithelial cells on day 5 (Supplementary Fig. 3Ba-c, white arrow), sparse and scattered signals on day 6 (Supplementary Fig. 3Bd-f), and complete absence by day 7 in wild-type embryos (Fig. S3Bg-i). Similarly, no P-H3 signal was detected in $Squid^{-/-}$ embryos at day 7 (Supplementary Fig. 3Bj-l). Immunohistochemistry further confirmed that Squid protein was expressed in wild-type midgut epithelial cells at day 7 (Supplementary Fig. 3Ca-c) and its absence in the mutant (Supplementary Fig. 3Cd-f).

Taken together, these findings demonstrate that Squid is essential for embryonic midgut development. The mutant exhibits lipid-like accumulation resembling the wild-type day-6 phenotype but lacks the corresponding proliferation signal and organized epithelial structure. This indicates that Squid may affect embryonic ISC differentiation rather than proliferation, underscoring its crucial role in the development and function of embryonic midgut epithelial cells.

### Knockout of *Squid* disrupts nutrition and energy supply during embryonic development

Lipids accumulated in eggs during oogenesis serve as the primary energy source for developing embryos[23], and the midgut is crucial for lipids digestion and nutrient absorption in insects[3]. To assess whether nutrient absorption is impeded in $Squid^{-/-}$ embryos, we performed Nile red staining on 7-day-old wild-type and $Squid^{-/-}$ embryos. The results showed no obvious lipid droplets accumulation in the midgut lumen of wild-type embryos (Fig. 3Aa-c), whereas numerous lipid droplets (yellow arrow) were present in the midgut lumen of $Squid^{-/-}$ embryos (Fig. 3Ad-f). Furthermore, comparative transcriptomic analysis of 7-day-old wild-type and $Squid^{-/-}$ embryos was conducted. Principal component analysis (PCA) revealed distinct separation between the groups along the first principal component (PC1), which accounted for 60.01% of the variance in differentially expressed transcripts (DETs) (Supplementary Fig. 4A). A total of 3,226 DETs were identified in $Squid^{-/-}$ embryos compared to wild-type, comprising 1,640 up-regulated DETs and 1,586 down-regulated DEGs (Supplementary Fig. 4B). Kyoto Encyclopedia of Genes and Genomes (KEGG) enrichment analysis indicated that down-regulated DEGs were enriched in pathways related to metabolic pathways, carbon metabolism, biosynthesis of amino acids, fructose and mannose metabolism, and citrate cycle (TCA cycle) (Fig. 3B), suggesting suppressed nutrient absorption and energy supply in the mutants.

Consistent with these findings, Real-time quantitative PCR (RT-qPCR) analysis revealed significant downregulation of all tested nutrient digestion- and absorption-related genes in $Squid^{-/-}$ embryos compared to wild-type (Fig. 3C–H). To examine lipid metabolism more directly, we measured the expression of key lipid-processing genes. RT-qPCR analysis showed that the expression levels of *lipase member H-A* were significantly

downregulated in $Squid^{-/-}$ embryos, whereas the expression levels of *fatty acid binding protein* (*fabp*) and *Apolipoprotein lipid transfer particle* (*Apoltp*) remained largely unchanged (Fig. 3I–K). Although lipid transport protein levels were unaffected, the reduction in *lipase* expression is expected to directly impair lipid hydrolysis, thereby limiting the release of absorbable free fatty acids.

Together, these results demonstrate that midgut digestive function and nutrient absorption capacity are compromised in $Squid^{-/-}$ embryos.

### Identification of Squid-regulated candidate genes during embryonic development

The hnRNP family comprises a diverse group of RNA-binding proteins that regulate the alternative splicing of various pre-mRNAs[24,25]. Squid, a conserved member of this family, regulates alternative splicing of specific genes in both *Drosophila* and *B. mori*[19,20,22]. To elucidate the molecular mechanisms underlying Squid's role in embryonic midgut development, we performed RNA-immunoprecipitation sequencing (RIP-seq) to identify its target transcripts. The effectiveness of immunoprecipitation was confirmed by the presence of Squid in both input and immunoprecipitated (IP) samples using an anti-Squid antibody, with no detectable signal in the IgG control (Fig. 4A). Subsequently, cDNA libraries constructed from IP and input mRNAs were subjected to Illumina sequencing. A total of 2,312 genes were identified as potential Squid targets based on significant enrichment in the IP group (|logFC|>1.0, *p*-value < 0.05) (Fig. 4B). Genomic distribution analysis revealed that the majority of Squid-binding sites were located within coding sequences (CDS) (Fig. 4C). Motif analysis using HOMER identified "GAAGGAA" as the predominant RNA motif bound by Squid (Fig. 4D).

Structural predictions via AlphaFold indicated that the RRM1 domain of Squid binds strongly to the "GAAGGAA" motif (ipTM=0.77), with key interactions mediated by specific residues including Lys44, Phe46, Leu50, Trp52, Lys75, Arg84, Phe88, Lys111, Asp114, Lys117, Lys118, Lys119, and Arg121 (Fig. 4E, F). Given the high evolutionary conservation of Squid RRMs across insects and mammals (Supplementary Fig. 1), these findings suggest that Squid may function through recognition of the "GAAGGAA" RNA motif in diverse species.

To further identify genes directly regulated by Squid, we analyzed RNA sequencing data from $Squid^{-/-}$ and wild-type embryos using replicate Multivariate Analysis of Transcript Splicing (rMATS) software to detect differential alternative splicing events. This analysis identified 294 genes exhibiting significant alternative splicing changes in $Squid^{-/-}$ mutants compared to wild-type embryos. Intersection of these genes with the 2,312 potential Squid-binding targets identified from RIP-seq revealed 72 high-confidence candidate genes whose alternative splicing is directly regulated by Squid during embryonic organogenesis (Fig. 4G). Gene ontology and pathway enrichment analysis indicated that these target genes were significantly associated with the Wnt signaling pathway and the AGE-RAGE signaling pathway in diabetic complications (Fig. 4H). Notably, *Axin* and *Smad4*, key regulators within the Wnt signaling pathway, were among the enriched genes (Fig. 4I). Given the well-established role of Wnt signaling in intestinal development, homeostasis, and regeneration in both mammals and *Drosophila*[26], these results implied that Squid may modulate embryonic midgut development through direct regulation of alternative splicing of *Axin* and *Smad4* within the Wnt signaling pathway.

### Squid regulates embryonic midgut development by fine-tuning β-catenin levels via *Axin* alternative splicing

To further investigate whether alternative splicing of *Axin* and *Smad4* was affected in $Squid^{-/-}$ embryos, the mRNA expression levels of *Axin* and *Smad4* isoforms were detected by RT-qPCR, respectively. Two *Axin* mRNA isoform exist: *Axin* isoform 1 (XM_012695811.3, *Axin-L*) and *Axin* isoform 2 (XM_021352163.2, *Axin-S*), with Axin-S lacking a 46-amino acid sequence present in Axin-L (Fig. 5A). RT-qPCR results showed that although total *Axin* mRNA levels remained unchanged between wild-type

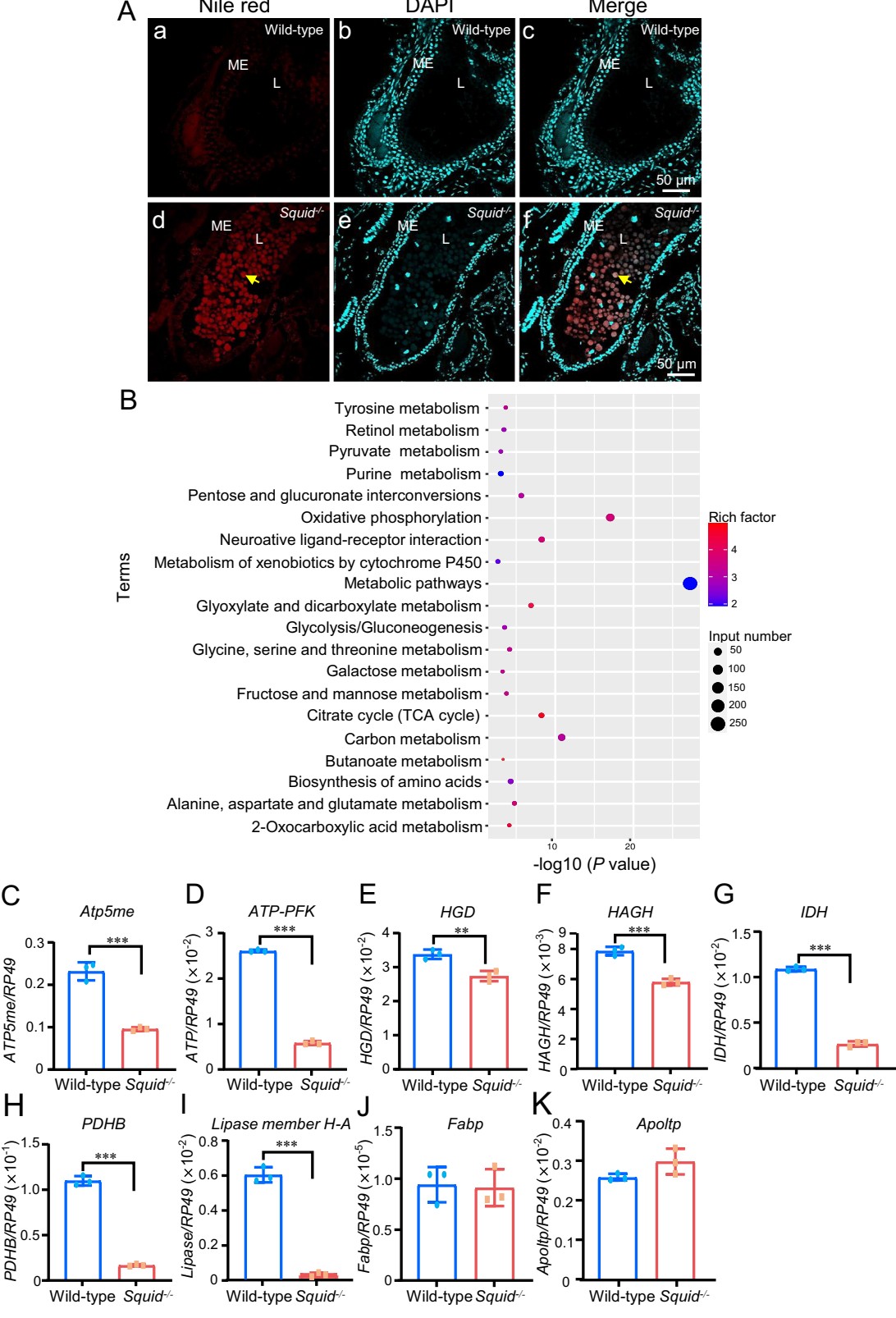

**Fig. 3 | *Squid* knockout disrupts nutrition and energy metabolism. A** Nile Red staining of lipid droplets in the midgut of wild-type and *Squid*$^{-/-}$ embryos at day 7 post-oviposition. Lipid droplets are indicated by yellow arrows. ME, midgut epithelium. L, lumen. Nuclei are marked by DAPI in cyan. The scale bars represent 50 μm. **B** KEGG pathway enrichment analysis of downregulated genes in *Squid*$^{-/-}$ embryos. **C–H** RT-qPCR validation of nutrient digestion and absorption-related genes in wild-type and *Squid*$^{-/-}$ embryos at day 7 post-oviposition. *Atp5me, H+ transporting ATP synthase subunit e. ATP-PFK, ATP-dependent 6-phosphofructokinase. HGD,* *Homogentisate 1,2-dioxygenase. HAGH, hydroxyacylglutathione hydrolase. IDH, Isocitrate dehydrogenase. PDHB,* pyruvate dehydrogenase E1 component beta subunit. The data are presented as mean ± SD. *n* = 3 biologically independent samples. **I–K** RT-qPCR validation of lipid digestion and absorption-related genes in wild-type and *Squid*$^{-/-}$ embryos at day 7 post-oviposition. *Fabp, fatty acid binding protein. Apoltp, Apolipoprotein lipid transfer particle (Apoltp).* The data are presented as mean ± SD. *n* = 3 biologically independent samples. Statistical significance was determined by using *t* test. *p* < 0.01 (**) and *p* < 0.001 (***).

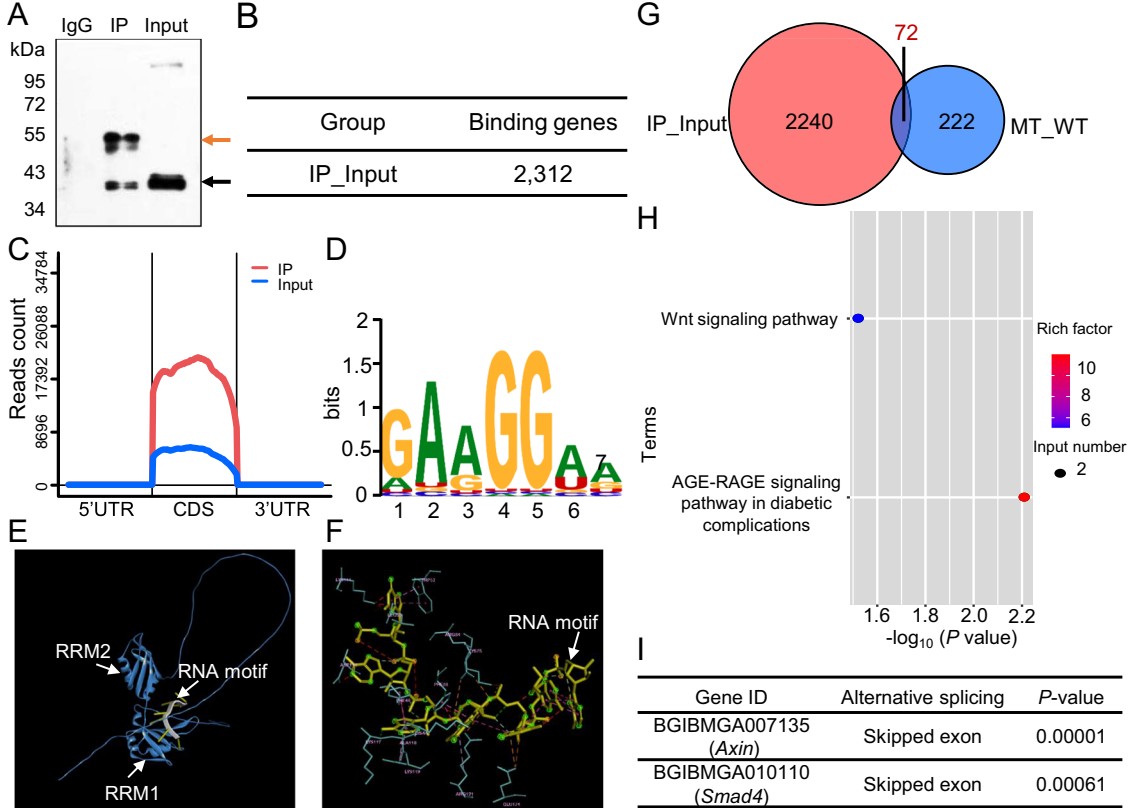

**Fig. 4 | Identification of potential target genes regulated by Squid. A** Validation of Squid immunoprecipitation in RIP assays using an anti-Squid antibody. Squid was pulled down using the anti-Squid antibody (lane 2, black arrow). IgG was used as a negative control and showed no pull-down of Squid (lane 1). The red arrow indicates the IgG heavy chain. **B** Numbers of genes identified as Squid putative binding genes. Genes with significant enrichment (|logFC| >1.0, *p*-value < 0.05) in the IP group compared to the input group were considered potential Squid-bound transcripts. **C** Genomic distribution of peaks enriched in Squid RIP-seq samples. **D** Top enriched RNA binding motifs for Squid, predicted using the Homer motif analysis tool.

**E** Structural prediction of Squid bound to its target RNA motif, modeled with AlphaFold. **F** Molecular surface representation of Squid showing amino acid residues involved in RNA motif binding, visualized using Discovery Studio. **G** Venn diagram illustrating the overlap between alternative splicing (AS) genes and Squid-bound genes. "IP-Input" denotes putative binding targets from RIP-seq. "MT-WT" indicates differentially alternatively spliced genes identified via RNA-seq in *Squid⁻/⁻* vs. wild-type embryos. **H** KEGG pathway enrichment analysis of Squid target genes identified through the Venn diagram overlap. **I** *Axin* and *Smad4*, components of the Wnt signaling pathway, were identified as direct targets of Squid.

and *Squid⁻/⁻* embryos, the expression of *Axin-L* was significantly reduced in mutants (Fig. 5B-C), suggesting a relative increase in *Axin-S*. These findings indicate that loss of Squid may promote alternative splicing favoring *Axin-S* production. For *Smad4*, which has three known isoforms, neither the total mRNA level nor the combined expression of *Smad4-1* and *Smad4-3* differed significantly between genotypes (Supplementary Fig. 5), indicating that *Smad4* splicing is not altered in *Squid⁻/⁻* embryos.

Axin is a known inhibitor of the Wnt signaling pathway, forming a complex with β-catenin, GSK-3β, and APC to promote β-catenin degradation[27,28]. We therefore examined β-catenin protein levels in *Squid⁻/⁻* embryos using an anti-β-catenin antibody raised against human β-catenin, which cross-reacts with *B. mori* β-catenin due to 70.8% sequence identity (Supplementary Fig. 6A, B). Western blot analysis revealed β-Catenin protein levels were markedly reduced in *Squid⁻/⁻* embryos compared to wild-type (Fig. 5D). Consistent with this, immunohistochemistry showed diminished β-catenin signal in the basal region of the midgut epithelium in 7-day-old *Squid⁻/⁻* embryos (Fig. 5E, F").

To functionally compare the *Axin* isoforms, we overexpressed Axin-L-EGFP and Axin-S-EGFP in *Bm*N cells and detected their effects on β-catenin content. Overexpression of Axin-L-EGFP significantly increased β-catenin levels, whereas Axin-S-EGFP showed no significant effect compared to the EGFP control (Fig. 5G, H), indicating distinct functional roles of the two isoforms in regulating β-catenin.

Together, these results suggest that Squid directly regulates alternative splicing of *Axin*, which is required to maintain physiological β-catenin expression levels in the basal midgut epithelium during embryonic development (Fig. 6).

## Discussion

The midgut serves as the primary site of digestion and nutrient absorption in insects[2,3], yet the molecular mechanisms governing its embryonic development remain incompletely understood. In this study, we identify the hnRNP family protein Squid as an essential regulator of embryonic midgut development in the lepidopteran model *B. mori*.

As a prominent member of the hnRNP family, Squid contains two conserved RRMs shared by insects and mammals (Supplementary Fig. 1). In *Drosophila*, it is known to regulate dorsal-ventral patterning, oocyte polarity establishment via mRNA localization, and maternal mRNA transcription during oogenesis[15–18,29]. Loss of Squid function results in embryonic lethality in *Drosophila*[30], and our findings show that depletion of *Squid* also causes embryonic death in *B. mori* (Fig. 1). Hence, Squid exemplifies an evolutionarily conserved hnRNP essential for insect embryogenesis.

In *Squid⁻/⁻* embryos, we observed severe morphological defects in midgut epithelium cells, including ISCs, CCs, and GCs (Fig. 2). Given that CCs are primarily responsible for digestive enzyme secretion and nutrient absorption[2,31], their structural disorganization implies a major functional impairment of the midgut. This is further corroborated by lipid droplet accumulation in the midgut lumen and broad suppression of nutrient metabolism pathways, as shown by transcriptomic and RT-qPCR analyses (Fig. 3 and Fig. 4). Collectively, these results establish

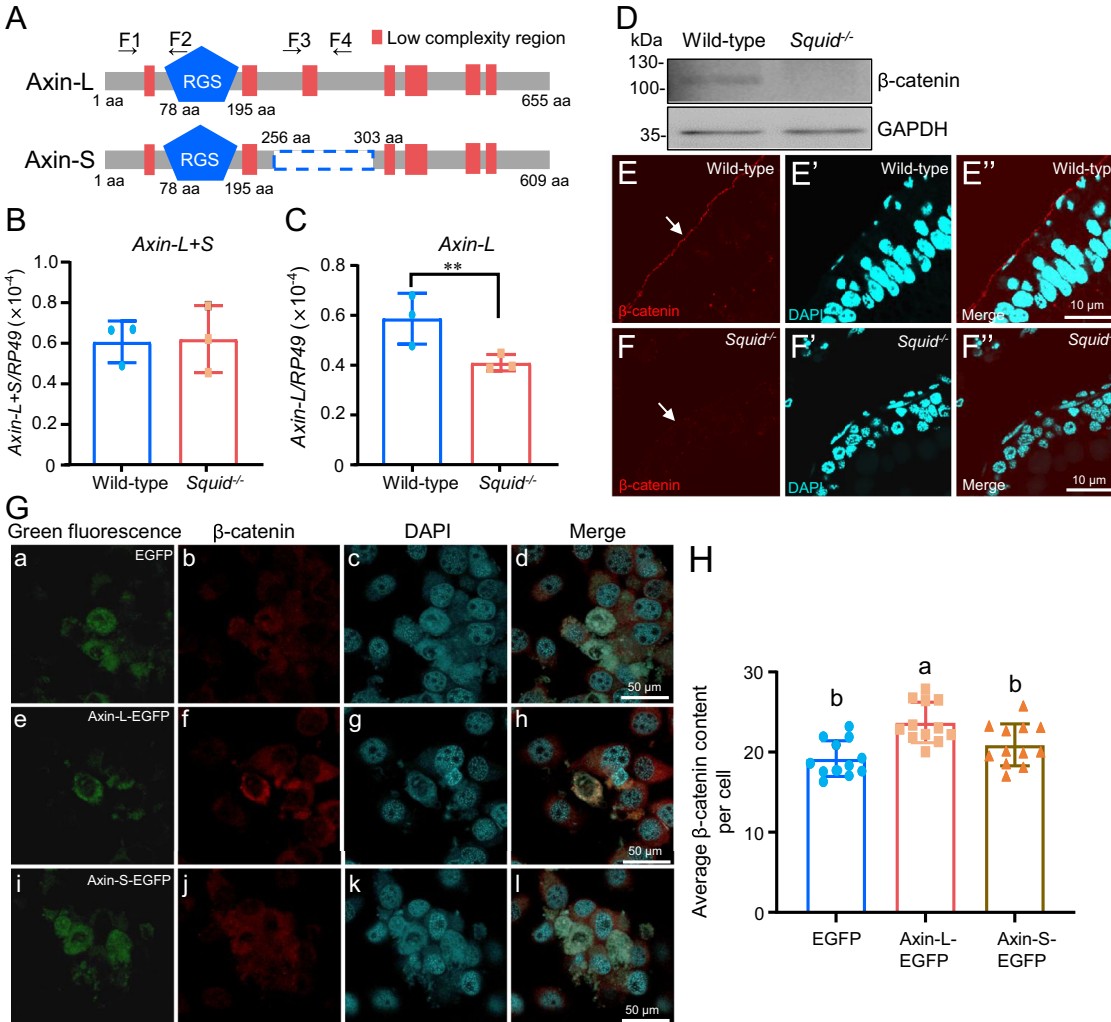

**Fig. 5 | Squid regulates embryonic midgut development through modulating** *Axin* **alternative splicing. A** Schematic representation of Axin isoforms Axin-L and Axin-S. The region absent in Axin-S is indicated by a blue dashed box. **B, C** RT-qPCR analysis of *Axin-L + S* (**B**) and *Axin-L* (**C**) in wild-type and *Squid^{−/−}* embryos. Primers F1 and F2 were used to amplify the shared region of the *Axin-L* and *Axin-S* mRNA, while Primers F3 and F4 were used to amplify *Axin-L* mRNA. The data are presented as mean ± SD. *n* = 3 biologically independent samples. **D** Western blot analysis of β-catenin levels in wild-type and *Squid^{−/−}* embryos at day 7. **E, F** Immunohistochemical detection of β-catenin levels in the midgut of wild-type and *Squid^{−/−}* embryos at day 7. Red fluorescent signals indicate β-catenin. Nuclei are marked by DAPI in cyan. The scale bars represent 10 μm. **G** Effect of overexpressing Axin-L or Axin-S on β-catenin levels in *Bm*N cells detected by immunohistochemistry. Red fluorescent signals indicate β-catenin protein. Green fluorescence indicates overexpressed EGFP (**a**), Axin-L-EGFP (**e**) or Axin-S-EGFP (**i**) protein. Nuclei are marked by DAPI in cyan. The scale bars represent 50 μm. **H** Quantified red fluorescence intensity of β-catenin per *Bm*N cell after overexpression of EGFP, Axin-L-EGFP or Axin-S-EGFP protein, respectively. The data are presented as mean ± SD. *n* = 14 biologically independent samples. Different letters above the bars denote statistically significant differences analyzed by two-way ANOVA.

Squid as indispensable for both proper morphogenesis and physiological function of the embryonic midgut.

To elucidate the molecular basis of these defects, we focused on the RNA-binding functions of Squid, which is known to regulate the alternative splicing of several pre-mRNAs, including *Hsrx-RC* and *Ddc* in *Drosophila*[19], as well as *POUM2* in *B. mori*[22]. RIP-seq identified 2,312 potential Squid target genes, of which 72 showed altered splicing patterns in *Squid^{−/−}* embryos. These alternative splicing targets were significantly enriched in the Wnt signaling pathway (Fig. 4). The canonical Wnt/β-catenin pathway, one of several Wnt-activated cascades, tightly controls β-catenin stability. In the absence of Wnt signaling, a destruction complex scaffolded by Axin promotes β-catenin degradation[32,33]. Wnt activation disrupts this complex, allowing β-catenin to accumulate, enter the nucleus, and drive target gene expression via Tcf/Lef transcription factors[34,35]. Beyond its transcriptional role, several evidence suggests that β-catenin also contributes to cell-cell adhesion independently of Wnt signaling by condensing with E-cadherin and α-catenin to promote junction assembly[36–38]. In *Squid* mutants, we observed markedly reduced β-catenin levels in the basal midgut epithelium, accompanied by differentiation arrested of ISCs (Fig. 2 and Supplementary Fig. 3). This correlation suggests that β-catenin deficiency may compromise cell-cell junctions, thereby disrupting epithelial integrity and the stem cell niche required for ISC differentiation.

We further revealed that Squid likely maintains physiological β-catenin levels during embryonic midgut development by regulating alternative splicing of *Axin*. Although total *Axin* mRNA is unchanged, the long isoform (*Axin-L*) is significantly reduced in mutants, implying a shift toward the short isoform (*Axin-S*) (Fig. 5). Consistent with reports that AXIN1 splice variants differentially modulate β-catenin levels in mammalian cells[39], overexpression of the long Axin isoform (Axin-L) in *Bm*N cells elevates β-catenin protein levels, whereas the short isoform (Axin-S) has no significant effect (Fig. 5G, H). These in vitro results indicate that Squid-dependent alternative splicing generates functionally distinct Axin isoforms with differential capacities to regulate β-catenin stability. An intriguing question arising from this finding is why the

**Fig. 6 | Schematic representation of Squid-mediated regulation of embryonic midgut development by modulating alternative splicing of _Axin_, thereby fine-tuning β-catenin levels.** ISCs intestinal stem cells.

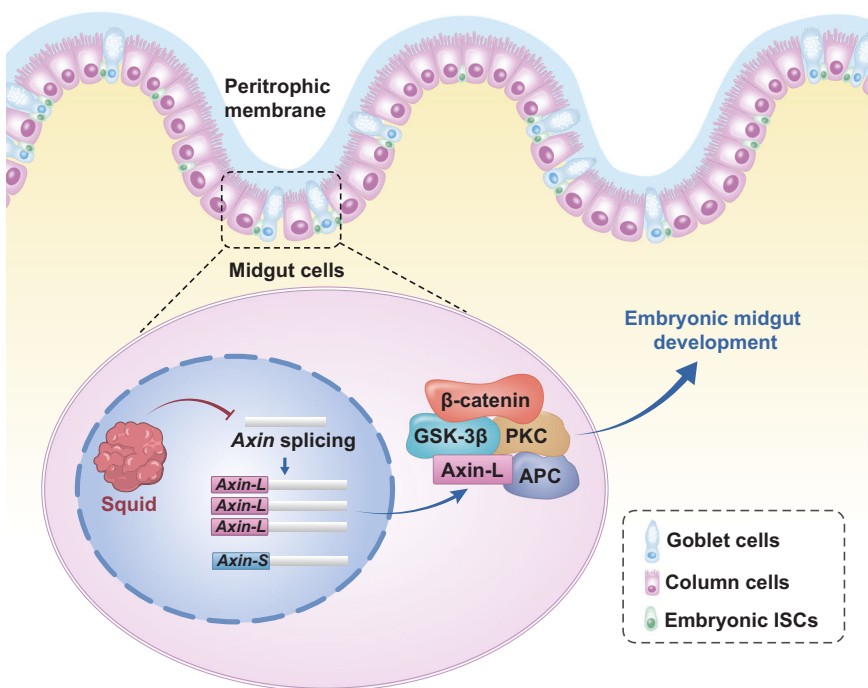

absence of a 46-amino acid low-complexity region (LCR) in Axin-S abolishes its effect on β-catenin levels (Fig. 5G, H). As LCRs can confer context-specific protein functions[40], this domain may be critical for Axin activity within the embryonic midgut microenvironment. Future in vivo studies in genetically tractable systems such as _Drosophila_ will be essential to validate the physiological relevance of the Squid-Axin-β-catenin axis and to define the precise role of the Axin LCR.

In summary, our findings reveal a previously unknown post-transcriptional regulatory layer in insect organogenesis. Our results show that the hnRNP protein Squid is essential for embryonic midgut development in _B. mori_. Specifically, Squid regulates alternative splicing of _Axin_, which in turn maintains normal β-catenin levels. Knockout of Squid may disrupt this pathway, leading to severe midgut developmental and functional defects that likely contribute to the embryonic lethality observed in _Squid_ mutants.

## Methods
### Experimental animals and cells
The silkworm strain P50 was supplied by the Research and Development Center of the Sericultural Research Institute of the Academy of Agricultural Sciences of Guangdong Province, China. The larvae were fed with fresh mulberry leaves and reared at 27 °C under a 12 h light/12 h dark photoperiod. _Bm_N (_B. mori_ ovaries derived) cell line was maintained at 28 °C in Grace's Insect Medium (Gibco, Grand Island, NY, USA) supplemented with 10% fetal bovine serum (FBS) (Gibco, Grand Island, NY, USA).

### CRISPR/Cas9-mediated knockout of _Squid_
The three 23-bp sgRNAs targeting the upstream region of _Squid_ were designed using the online platform CRISPRdirect (http://crispr.jp/) and amplified by PCR with an Ex Taq® kit (TaKaRa Co., Dalian, China). The sgRNA sequences were listed in Table S1. In vitro transcription of sgRNAs was performed using the MAXIscript T7 Kit (Ambion, Austin, TX, USA) according to the manufacturer's instruction. Briefly, a DNA oligonucleotide template containing the T7 promoter sequence followed by the 23 bp sgRNAs sequence was designed and amplified by PCR, then purified PCR product was used as the template for in vitro transcription. The transcription reaction was prepared in a 20 μL volume containing 1× reaction buffer, 1 μL each of ATP, CTP, GTP, and UTP, 1 μL T7 RNA polymerase, and 1 μg

of DNA template. The reaction was incubated at 37 °C for 6 h. After transcription, residual DNA template was removed by digestion with the kit's DNase I (2 U, 37 °C, 15 min). After digestion, the sgRNA was purified by ethanol precipitation, quantified, and stored at -80 °C for subsequent use.

A mixture of 3 μL Cas9 protein (PNA BIO, Thousand Oaks, USA) and 5 μL pooled sgRNAs (1200 ng/μL) was microinjected (8 nL/egg) into freshly laid silkworm eggs (within 2 h post-oviposition) using a microinjector (Eppendorf FemtoJet 4i, Hamburg, Germany). The injected eggs were incubated at 27 °C under a 12 h light/12 h dark photoperiod until hatching.

In the G0 generation, 300 eggs were injected, resulting in a hatching rate of 17%. A total of 10 adults were obtained, among which only one was confirmed as a mutant. For genotyping, genomic DNA was non-invasively extracted from pre-pupal exuviae and used for PCR amplification and sequencing of the _Squid_ locus[41]. Heterozygous mutants (_Squid_[+/−]) displayed normal growth, development, and reproductive capacity, allowing routine screening via exuviae analysis. In contrast, homozygous mutants (_Squid_[−/−]) exhibited significantly delayed embryonic development at 7-8 days post-oviposition. While wild-type and heterozygous embryos developed head pigmentation by day 7, homozygous mutants lacked this pigmentation (Fig. 1C-D). Developmentally delayed eggs were collected for _Squid_ genotyping by sequencing and used in subsequent experiments.

### Western blot
The polyclonal anti-Squid antibody was generated and its specificity was extensively validated, as detailed in our previous publication[42]. For protein extraction, wild-type and _Squid_[−/−] eggs were homogenized in RIPA Complete Lysis Buffer (Beyotime, P0038-100 mL) and centrifuged at 12,000 × g for 20 min at 4 °C. The supernatant was filtered through 0.45 μm membranes and stored at −80 °C. Protein concentration was determined using the Bradford method[43]. A total of 30 μg of protein extracted from day 7 embryos was mixed with 5× loading buffer (250 mM Tris-HCl, pH 6.8, 10% SDS, 0.5% bromophenol blue, 50% glycerol, 5% 2-mercaptoethanol) and separated on 10% SDS-polyacrylamide gels. Subsequently, proteins were transferred to polyvinylidene fluoride (PVDF) membranes. The membranes were blocked overnight at 4 °C with 3% (w/v) bovine serum albumin (BSA) in Tris-buffered saline containing 0.1% Tween 20 (TBST). After three 10-minute washes with TBST, the membranes were incubated with gentle shaking at 37 °C for 2 h with either anti-Squid antibody (1:2000 dilution) or

anti-β-catenin (1:1000 dilution). Following another series of washes, the membranes were incubated with horseradish peroxidase (HRP)-conjugated secondary antibody (1:10,000 dilution) under the same conditions. Mouse anti-GAPDH (1:5000 dilution, Proteintech, 60004-1-Ig) or anti-β-tubulin (1:5000 dilution, Dingguo Changsheng, TB002-M) antibodies were used as internal controls.

## HE staining and immunohistochemistry

Wild-type and *Squid*⁻/⁻ eggs were collected and fixed overnight in 4% paraformaldehyde. The embryos were subsequently dehydrated through a graded series of ethanol and xylene, and embedded in paraffin. Sections of 5 μm thickness were prepared using an ultramicrotome (EM UC7, Leica, Wetzlar, Germany). The sections were mounted on glass slides, deparaffinized in xylene and rehydrated through a descending ethanol gradient in preparation for HE staining and immunohistochemistry. HE staining was performed using a Servicebio® HE dye solution kit (Servicebio, Wuhan, China). Briefly, sections were stained with hematoxylin for 4 min, rinsed, differentiated briefly (5 s), rinsed again, and blued (5 s). After thorough rinsing, samples were dehydrated through 85% and 95% ethanol (5 min each), stained with alcohol-based eosin for 5 min, dehydrated in absolute ethanol (twice, 5 min each), cleared in xylene (twice, 5 min each), and finally mounted using neutral resin.

For immunohistochemistry, after permeabilization with proteinase K (10 μg/mL) for 10 min, the sections were blocked with 3% BSA in PBS at 37 °C for 1 h to prevent non-specific binding. They were then incubated overnight at 4 °C with the primary antibodies (anti-Squid, 1:200 dilution; anti-β-catenin (1:200 dilution, Immunoway, YM3403); Histone H3 (Phospho Ser10, P-H3) (1:200 dilution, Immunoway, YP0129). Following primary antibody incubation, the sections were treated with a secondary antibody (Alexa Fluor 594, 1:400 dilution, Invitrogen, USA) at 37 °C for 2 hours in the dark. Nuclei were counterstained with DAPI. Fluorescence images were acquired using an FV3000 confocal microscope (Olympus, Tokyo, Japan).

## Morphological and structural observation by TEM

Wild-type and *Squid*⁻/⁻ eggs at day 7 post-oviposition were collected, dechorionated, and fixed overnight at 4 °C in 2.5% glutaraldehyde. Subsequent processing and TEM analyses were performed as previously described[22]. Fixed samples were washed four times with 0.1 M phosphate buffer (15 min each), post-fixed in 1% $OsO_4$ for 1.5 h, and dehydrated in an ethanol series (30, 50, 70, 80, 90, and 100%, 15 min per step). After dissecting the embryonic body, tissues were infiltrated with Spurr's resin using an acetone: resin gradient (3:1 for 2 h, 1:1 for 3.5 h, 1:3 overnight, then pure resin for 7.5 h). Following vacuum-assisted infiltration, tissues were embedded in gelatin capsules and polymerized at 70 °C for 12 h. Ultrathin sections (~70 nm) were cut using an ultramicrotome (EM UC7, Leica, Wetzlar, Germany) and stained with uranyl acetate and lead citrate prior to imaging with a Thermo Fisher Talos L120C transmission electron microscope.

## RNA sequencing (RNA-Seq) and transcriptomic analysis

Total RNA was isolated from wild-type and *Squid*⁻/⁻ embryos collected on day 7 post-oviposition. RNA sequencing libraries were constructed using the NEBNext® Ultra™ RNA Library Prep Kit for Illumina® (NEB, Beijing, China) according to the manufacturer's protocol. Briefly, total RNA was purified, fragmented and reverse-transcribed into first-strand cDNA, followed by second-strand synthesis. The double-stranded cDNA was end-repaired, ligated to adaptors. The ligated products were size-selected, PCR-amplified with index primers, PCR product purification, and library quality evaluation. The libraries were sequenced on the NovaSeq X Plus platform (Illumina) by Majorbio Company (Shanghai, China). The amount of clean bases for all samples exceeded 7 G (Gigabases, G), with Q30 scores above 90%, and the alignment rates of the sequencing data to the reference genome reached approximately 90%. These results indicate the sequencing quality was high, and the data are suitable for subsequent analyses.

Raw reads were subjected to quality control and filtering, then aligned to the reference silkworm genome (http://silkworm.genomics.org.cn/) using TopHat v2.0.12[44]. Read counts per gene were obtained using HTSeq v0.6.1. Gene expression levels were estimated as FPKM (fragments per kilobase of transcript per million mapped reads) values. Novel transcripts were identified with Cufflinks v2.1.1[45]. DEGs were identified with DESeq2, using thresholds of |logFC| > 1 and $p < 0.05$. Differential alternative splicing events between *Squid*⁻/⁻ and wild-type embryos were detected using replicate rMATS software[46].

## RIP-seq

RIP-seq was performed to identify the genes bound by Squid in silkworm embryos at the organogenesis stage (76 h post oviposition). Embryos were homogenized and lysed in RIPA Complete Lysis Buffer (Beyotime, P0038-100 mL). From the resulting lysate, 10% of the supernatant was set aside as input control. The remaining supernatant was subjected to immunoprecipitation by incubation with either anti-Squid antibody or normal IgG (negative control) overnight at 4 °C, followed by incubation with Protein A/G Magnetic Beads (Invitrogen, USA) for 1 h at 4 °C. The input and immunoprecipitated RNAs were extracted using TRIzol reagent (Invitrogen, California, USA) and sequenced on Novaseq 6000 sequencer with PE150 model by SeqHealth (Wuhan, China). Sequencing reads were aligned to silkworm genomics (http://silkworm.genomics.org.cn/). The amount of clean bases for all samples exceeded 2 G, with Q30 scores above 90%, and the alignment rates of the sequencing data to the reference genome surpassing 90%. The RSeQC (version 2.6) was used for reads distribution analysis[47]. Peaks were annotated using bedtools (Version 2.25.0)[48]. The differentially binding peaks were identified by a python script, using fisher test[49]. DEGs were identified with edgeR (version 3.12.1), using thresholds of |logFC| >1.0, $p$-value < 0.05. Sequence motifs enriched in peak regions were identified using Homer (version 4.10)[50].

## RT-qPCR

Total RNA was extracted from day 7 wild-type and *Squid*⁻/⁻ embryos using TRIzol™ Reagent (Life Technologies, Carlsbad, USA). First-strand cDNA was synthesized using M-MLV Reverse Transcriptase (TaKaRa Co., Dalian, China). RT-qPCR was performed using the Eastep® qPCR Master Mix Kit (Promega, Shanghai, China). Reactions were assembled in a total volume of 20 μL, containing 10 μL of 2 × qPCR Master Mix, 0.4 μL each of forward and reverse primers (10 μM), 2 μL of template cDNA, and 7.2 μL of nuclease-free water. The reactions were run on a QuantStudio™ 6 Flex (Thermo Fisher Scientific, Waltham, USA) under the following conditions: 95 °C for 2 min, followed by 40 cycles of 95 °C for 15 s and 60 °C for 1 min. Gene expression levels were quantified using the comparativeCt ($2^{-\Delta Ct}$) method, with *RP49* (Accession number: NM_001098282) serving as the internal reference gene. The sequences of all primers used are provided in Supplementary Table 1.

## Structural modeling and functional prediction

The three-dimensional structure model of Squid was predicted using SWISS-MODEL (https://swissmodel.expasy.org/). The interaction between Squid and target RNA motifs was predicted using AlphaFold3[51]. Key amino acid residues involved in RNA binding were visualized and analyzed using Discovery Studio.

## Effect of overexpression of Axin-L and Axin-S in *Bm*N cells on β-catenin levels

The coding sequences of *Axin-L* or *Axin-S* were cloned into pIE1-EGFP vector to generate two expression vectors: p*Axin-L*-EGFP and p*Axin-S*-EGFP. *Bm*N cells were transfected with pEGFP, p*Axin-L*-EGFP, and p*Axin-S*-EGFP, respectively, using FuGENE HD reagent (E2311, Promega) and the cells were fixed in 4% paraformaldehyde for 30 min, washed in PBS, and stained with an anti-β-catenin antibody (1:200 dilution) and DAPI post 48 h transfection. Images were acquired using an Olympus Fluoview FV3000 confocal microscope. The quantification of red fluorescence intensity in

*Bm*N cells was conducted by Image J. Mean fluorescence intensity was measured using the ROI created.

## Statistics and Reproducibility

All experiments included at least 3 individual biological repeats. Statistical analysis of the data was performed using Student's *t*-test or two-way ANOVA. For the *t*-test: $*p < 0.05$; $**p < 0.01$; $***p < 0.001$. The data are presented as mean ± SD. The statistic data of all images are represented in Supplementary Data File.

## Reporting summary

Further information on research design is available in the Nature Portfolio Reporting Summary linked to this article.

## Data availability

All data required to evaluate the conclusions drawn in this paper are presented within the main text and the supplementary materials. The reads of transcriptome and RIP-Seq were deposited in the Sequence Read Archive (SRA) database of NCBI with the accession number SRP542796 and SRP543009, respectively. The numerical source data can be found in the Supplementary Data File. The uncropped blot images can be found in the Supplementary Figs. (Supplementary Fig. 7 and Supplementary Fig. 8). The data and materials are also available as requested directly from the corresponding authors. Data Sources are provided with this paper.

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

## Acknowledgements
We thank Chuanhe Liu of South China Agricultural University for her help in sample preparation for transmission electron microscopy. This study was supported by grants from the National Natural Science Foundation of China (No. 31872969).

## Author contributions
H.D. designed and monitored the project. C.T. and W.M. performed the experiments. J.H. performed the TEM experiment. C.T. interpreted the data. M.C. and Y.P. prepared some of the experimental materials. H.D. and C.T. wrote the manuscript. K.L. performed P-H3 staining experiment and revised the manuscript.

## Competing interests
The authors declare no competing interests.
