## [Transparent Peer Review file · Communications Biology]

The RNA-binding protein Squid regulates embryonic midgut development via Axin alternative splicing in *Bombyx mori*

Corresponding Author: Professor Huimin Deng

Version 0:

Reviewer comments:

Reviewer #1

(Remarks to the Author)

Tong et al. present a study of the role of Squid in *Bombyx mori*, the silkworm. They find a role in midgut development during embryogenesis, which they characterize morphologically. They then show that Squid regulates the alternate splicing of Axin, potentially tying Wnt signaling to the morphological and genetic changes in midgut development seen in Squid mutants.

Unfortunately, although well carried out technically, Justification for the study and support for the broad conclusions was not strong. Squid is likely ubiquitously expressed in *Bombyx*. Why focus on midgut? Although it is expressed there, it is likely expressed elsewhere as well. Similarly, why use *Bombyx*? Is there something special about this insect that drives the study of Squid in this species? There are many reasons to study diverse animal models but is there a specific reason to study Squid in the midgut in this species? Is Squid expressed and/or active in the *Drosophila* midgut? Finally, as described below, the link between Axin alternative splicing and the defects seen in midgut morphology are correlative as presented here. Additional work would need to be done to test whether or not these change in Axin are causal.

Specific comments

Figure 1 - *B. mori* Squid CRISPR appears to have worked well and the large decrease in Squid protein is convincing. However, more information is needed about the process – number of embryos injected, number surviving to various stages, including adult, and number of mutants obtained. Most importantly, it seems as if just one single heterozygote was recovered, suggesting that all animals studied result from a single event in one animal. This is supported by sequence showing the same 2 bp deletion in all sequences. This may be problematic as it is essential to ensure that there is no other mutation in this line that could contribute to the mutant phenotype. Having multiple mutant lines, with different mutations that resulted from different events, ideally from cleavage by different gRNAs would provide confidence that the Squid mutation is causal. If not that, at least 2-3 generations of backcrosses should be done. Ideally, rescue by wild type Squid would be definitive evidence, although transgenesis would be required and may be problematic in *Bombyx*.

Pg. 9: the authors justify looking at midgut because they previous showed a role for Squid in the cuticle. I don't understand the logic

Figure 2 – defects in the midgut seem convincing, although that is not the expertise of this reviewer

Figure 3: Why was RNA-seq carried out on whole embryos rather than dissected midguts? Also, what stage were these embryos? These choices need to be explained and justified since Squid is likely involved in many different processes. Bottom of page 10 states that the KEGG analysis points to the midgut. But many of the processes highlighted take place in many different cells and tissues of the animal. This conclusion needs to be critically analyzed and/or better explained.

Figure 4: (1) As for the RNA-seq, RIP-seq should have been carried out on dissected midguts at the stage of development when Squid would be expected to be active. The authors state that “embryos in organogenesis” were used. It is not clear what that means or what/why this stage was chosen. On the technical side, this approach seems to have been well executed and carefully analyzed. (2) The binding to coding sequences is unexpected. Were unspliced, full length RNAs recovered in the RIP-seq? Would binding to introns, splice borders and also 3'UTRs not be expected for a general RNA binding protein? (3) Genes differentially spliced are shown to fall into two major pathways (Wnt and AGE-RAGE). How many of the 72 genes were in each of these pathways? Were Axin and Smad4 the only Wnt-pathway genes found? If not, why were these chosen

for further analysis? Are they just good candidates or was there other evidence?

Figure 5: The data on changes in Axin isoforms and b-catenin levels in the Squid mutant are convincing. However, it is unclear how this alternate splicing event would impact b-catenin levels. What is the function of the alternatively spliced region? Is Axin causal here or is some other Wnt pathway gene involved? B-Catenin itself? Finally, the RT-PCR was done on whole embryos and should have been done on midgut. Thus, two aspects of conclusions such as this one on page 15 are not fully justified "Our study reveals that Squid is involved in the Wnt signaling pathway by regulating the alternative splicing of Axin during embryonic midgut development" – (1) there is no evidence that the alternative splicing of Axin causes the changes in b-catenin levels and (2) there is no evidence that the alternative splicing of Axin occurs in the midgut.

Finally, although embryonic lethality is clear and convincing, the midgut changes are not necessarily the cause of lethality, as claimed, for example, in the last sentence of the Discussion. Many other embryonic processes are likely implicated, as evidenced by the large number of change in the RNA-seq and RIP-seq experiments done on whole embryos.

Minor Points

There are small grammar mistakes in various places in the manuscript (eg, noun-verb agreement). These mistakes are quite minor and do not hamper understanding.

Yellow dots and many of the letters are hard to see in Figure 2

Figure 5E – b-catenin cannot be seen in merge (E'')

Response to specific questions

- Does the manuscript have technical or conceptual flaws that should prohibit its publication? There are no technical flaws. The experiments are well-designed and carefully carried out from a technical point of view.
- Are the conclusions original? The conclusions are original but not strongly supported by the data.
- Do you feel that the results presented are of immediate relevance for people in your own discipline or for a broader audience? I feel this result is incremental and the authors have not justified or explained the broad interest.

Reviewer #2

(Remarks to the Author)

In this study the authors investigated the role of Squid, a member of hnRNPs family, in midgut development in the silkworm, *Bombyx mori*. After demonstrating that Squid knockout impairs midgut formation during embryogenesis, they identified target genes and demonstrated that one of them (Axis) undergoes alternative splicing mediated by Squid.

As indicated in the article, although insect midgut has been widely analyzed from different points of view, the molecular mechanisms regulating its development are scarcely known. Thus, novel information on this topic is welcome, especially if it has been obtained in a model insect as the silkworm. However, I think that the manuscript presents many flaws and, in particular, five key aspects need to be carefully addressed:

1. The quality of some images is poor and it's thus difficult to finely evaluate the effects of Squid knockout on the midgut. In particular, some images cannot be compared (wt VS Squid $-/-$) and additional pictures must be provided to substantiate the authors' conclusions as detailed below.
2. Although this represents a key aspect of the study, the authors did not provide any direct experimental evidence demonstrating the impairment of nutrient digestion and absorption after Squid knockout, except showing a variation in the amount of lipid droplets in the midgut lumen. Moreover, also intestinal stem cells seem to be affected by impairing Squid signaling but, once again, no specific experiments have been performed to this purpose. This lack of data significantly affects the characterization of the role of Squid in this setting and, consequently, reduces the impact of the study.
3. Along the manuscript, the authors draw a parallelism between i) the silkworm and *Drosophila*, and ii) the silkworm and mammals. It is not completely clear to me if this is a pioneering study and these regulatory mechanisms have been here characterized for the first time in the silkworm (this would provide a significant novelty to the current study) or the present study aims at verifying the existence of homologous mechanisms in this insect starting from previous evidence obtained in the fruitfly. In relation to the second aspect indicated above, I did not understand if this study aims at paving the way to future studies in mammalian models. In this case, please consider that the insect midgut and the human digestive system are significantly different.
4. Materials and methods. Some additional information is necessary to reproduce the experiments.
5. Discussion must be significantly improved. Currently this section of the manuscript contains a summary of the main results integrated with some papers taken from the literature. However, the cited literature is sometimes inconsistent, not appropriately linked and does not contribute to develop a critical Discussion. Please note that it is not necessary to mention figures in this section.

- P3 L10. This reference does not directly support the point being made.

- Paragraph 2.2. How did they perform the screening of insects and the characterization of the obtained lines?
- P5 L23-24. What do they mean with "sections were affixed to the slides"?
- Paragraph 2.3. Why did they use proteinase K on tissue sections? Moreover, information on the two antibodies (source, concentration of use, etc.) must be given.
- P6 L8. How were proteins isolated from the tissue?
- P7 L12. Please provide information on the lysis buffer.
- P8 L3. Is RP49 adequate as housekeeping gene in this setting? Is its expression stable along development?
- P8 L14. Do they mean in *Bombyx mori*?
- P8 L26. Please note that in Fig. S2C the homozygote is reported.
- P9 L15 (and Figs. 2A-C). It is difficult to interpret these images and the insect anatomy is quite strange. Why is the midgut so close to epidermis? Moreover, I do not see the fat body.
- P9 L16-19. The correlation of this previous evidence taken from the literature with the current paragraph is not clear.
- P9 L19 to P10 L3 (and Figs. 2D-E). How can the authors be sure that these are lipids? Did they perform any histochemical staining or biochemical analysis? Moreover, both figures are difficult to be interpreted (see lumen, microvilli, spherites, arrows, etc.). Finally, variation in stem cells must be clearly visible and specific (proliferation) assays should be used to this purpose.
- P9 L24-25. Please consider that the midgut epithelium also contains endocrine cells.
- P10 L12-14. Although the authors here declare that they performed transcriptomic analysis to demonstrate that Squid depletion impairs nutrient absorption, they do not provide any evidence showing that the expression of transcripts involved in nutrient digestion and absorption are modified in this organ. Moreover, they conclude that their data indicate that the depletion of Squid leads to abnormal embryonic midgut development, ultimately inhibiting overall embryonic development (P10 L23-25). I'm quite confused about this final statement.
- P13 L1-2 (and Figs. 5E-F). The fluorescent signal shown in these figures does not allow to assess the precise localization of B-catenin in midgut samples. Moreover, did the authors hypothesize any cytoplasm-to-nucleus translocation of B-catenin as indicated in the Discussion? Finally, the figures need labels and orientation should be reconsidered.
- Figure 6. Please indicate stem cells as "ISCs".
- Figures. Scale bars must be revised (see font dimension and orientation).
- The paper needs a language revision by a mother tongue or a professional editing service. There is a misuse of some terms and the paper contains many typos.

Reviewer #3

(Remarks to the Author)

In the current manuscript, Tong et al. examine molecular function of a heterogeneous nuclear ribonucleoprotein, Squid, in embryonic development in the silkworm, *Bombyx mori*. The authors previously demonstrated that a target gene of Squid is responsible for cuticular development during embryogenesis (Tong et al., 2023). In this manuscript, the authors generated a Squid gene knockout (KO) silkworm by CRISPR-based mutagenesis. The Squid KO silkworms died during embryogenesis with defects in midgut development. In addition, the authors conducted transcriptome analysis and RNA-immunoprecipitation sequencing analysis to determine target genes of Squid that controls embryogenesis. The authors identified that Squid regulates alternative splicing of Axin, a Wnt signaling regulator. The authors performed multiple analyses to demonstrate molecular function of Squid. In particular, they determined the binding motif of Squid and the target genes of Squid. However, the current manuscript contains critical errors that makes it impossible to interpret the data properly.

Major points:

1) The major concern is the lack of evidence that Squid within the midgut regulates the midgut development during embryogenesis. As shown in Figure 1C and D, the embryonic development of the Squid KO silkworm is already delayed or arrested before day-7 after egg laying. The authors then compare the midgut development of 7-day-old embryos between the wild-type and Squid KO silkworms. Their developmental stages are completely different at that time. Therefore, it is likely that the midgut of the Squid KO silkworm is not disordered as the authors describe (p.9, l. 21) but rather underdeveloped. Considering the high expression of Squid during early embryogenesis, as shown by the authors in the previous study (Tong et al., 2023), Squid is likely required for the early embryogenesis. Moreover, it is unclear whether Squid is specifically expressed in the midgut. The authors emphasize the expression of Squid in the midgut and epidermal cells (p. 9, ll. 14-15). However, Figure 2A-C shows that Squid is detected in most cells. The authors need to provide evidence showing that Squid regulates midgut development within the midgut cells.

2) There are serious errors in the analysis and conclusions of β -catenin. Considering the function of Axin as a negative regulator of β -catenin, Axin has weak activity in the wild type at Day-7, suggesting that Axin-1 does not function as a Wnt signaling regulator, but Axin-2 does. This is unlikely, as Axin-1 contains functional domains (Figure 5A). If this interpretation is correct, the amount and localization of β -catenin is not regulated by Axin, but rather controlled by other factors such as the Wnt ligand. Moreover, as described above, developmental stages of the wild-type and Squid KO silkworms are different. Wnt signaling fluctuates along with embryonic development (Mundaca-Escobar et al., Front. Cell Dev. Biol., 2022). Therefore, the authors should compare these at the same developmental stage.

3) The source information for the anti-Squid polyclonal antibody is missing in the manuscript. The authors need to cite relevant articles if the antibody has already been used in previous studies. Otherwise, the authors need to indicate how to obtain the antibody. It is unclear whether the antibody specifically binds to Squid. Although the authors show that the Squid KO silkworm does not have the target protein (Figure 1F), they do not show the molecular size of bands or the immunocross-reactivity of the anti-Squid antibody. The authors should show the entire blot with molecular markers, as well as a control

lane with the second-antibody only, if this is the first study to use this antibody. The authors should not use this antibody for immunohistochemistry (Figure 2A-C), if it cross-reacts with other target proteins. It is also confusing that there are two bands in Figure 1F. Although the authors mention that these are Squid-1 and Squid-2 (p. 9, ll. 6-7), they did not explain about Squid-2.

4) The authors describe accumulation of lipid droplets in the midgut lumen in the Squid KO silkworm (Figure 2E). To the best of my knowledge, however, lipids cannot be stained by HE staining.

Other points should be corrected for the future submission:

- The sizes of the images in Figure 2D' and Figure 2E' are different. Figure 2E' may be twice as large as Figure 2D'. Also, horizontal scale bars should be added to all images.
- For immunohistochemistry, authors treated samples with Proteinase K (p. 6, l. 1). This treatment digests target proteins.
- The authors used 30 µg of proteins for Western blotting (p. 6, l. 8). The methods for protein extraction and quantification should be provided.
- The authors use the TopHat software for RNA sequencing. The TopHat developers recommend to use Hisat2 instead of TopHat (<https://ccb.jhu.edu/software/tophat/index.shtml>), as of low accuracy of TopHat.
- Add reference articles of the RIP-seq software if any.
- The authors analyzed differentially binding peaks by the "python script" (p. 7, ll. 21-22). The detail should be provided. If it is on a public data base, that should be cited. If it is made by the authors, the script should be provided.
- According to the Figure S2A, the authors prepared 3 sgRNAs. However, mutation was observed at the specific locus on the genome (targeted by S3?). Did authors use a mixture of sgRNAs, or was one sgRNA used for each egg? This point should be explained clearly.

Version 1:

Reviewer comments:

Reviewer #2

(Remarks to the Author)

This is the revised version of a manuscript in which the role of Squid in silkworm midgut development was investigated. The paper has been largely revised and significantly improved. The authors added new experimental evidence to corroborate their results and revised the text detailing several unclear aspects and providing more information on the experimental approach. Thus, the overall quality of the product has been improved. However, I'm still concerned about the impairment of nutrient digestion and absorption after Squid knockout. I understood the authors' explanation on this issue but, although they provided qPCR results showing the reduced expression of some genes, these are mostly coding for factors involved in metabolism (glycolysis, pyruvate metabolism, etc.), not directly linked to nutrient digestion and absorption. Thus, I'm still wondering if there is a real effect on digestive and absorption properties of this organ after Squid knockout, and I think that this aspect needs to be carefully considered before publication.

Reviewer #3

(Remarks to the Author)

In the revised manuscript, the authors provide new data to address my previous comments. Some issues, such as insufficient methodological descriptions, have been clearly resolved. However, revised manuscript does not address the critical problems I pointed out in the previous comment. Particularly, the evidence supporting the claim that Squid within the midgut cells contributes directly to midgut development remains weak. The authors have not ruled out the possibility that Squid indirectly regulates midgut development, as described below.

1) As I commented in my previous review, the authors have not resolved the issue that embryonic development in Squid knockout (KO) is delayed or arrested. Because of this, it remains possible that the abnormal midgut structure observed in Squid KO embryos is a consequence of entire developmental delay or arrest, rather than a midgut-specific developmental defect caused by the absence of Squid.

Although the authors state that midgut cell proliferation is completed by 7-day-old eggs in both wildtype (WT) and Squid KO embryos, based on phospho-HistoneH3 observation (Fig. S4, pp. 12-13, ll. 16-1), they do not provide the precise developmental timing at which midgut cell proliferation is completed in WT. Therefore, it remains possible that embryonic development in Squid KO is delayed or arrested after the midgut cell proliferation stage. The authors should perform positive control assays for phospho-histoneH3 detection at earlier developmental stages (e.g. Day-4 to Day-6 embryos), rather than in larval wing disks. I also strongly recommend analyzing midgut development at

earlier stages, at least in WT embryos, to demonstrate that the abnormal midgut structure in Squid KO (abnormal cell morphology and lipid accumulation) is distinct from that in early developmental stages in WT. Ideally, time-lapse analysis should be performed in Squid KO embryos, however, I understand that collecting homozygotic mutants prior to pigmentation may be challenging.

2) The role of Wnt signaling regulated by Squid and Axin in midgut development is unclear. In Figure 5E, β -catenin is localized to the cell membrane rather than the nucleus. β -catenin is known to localize to the membrane in the absence of Wnt signaling (Valenta et al., EMBO J., 2012). Thus, high abundance of β -catenin does not necessarily indicate that Wnt signaling is activated in WT embryo to induce cellular differentiation.

The authors provide no evidence that 1) Squid-mediated increases in Axin-L enhance Wnt signaling, or 2) enhanced Wnt signaling contributes to midgut cellular differentiation. Furthermore, as noted by all reviewers, the authors did not provide any evidence that Squid-mediated increases in Axin-L occur within the midgut. Therefore, the relationship between Squid and Wnt signaling should be discussed more carefully. For example, this study does not "demonstrate" that Squid contributes to the Wnt/ β -catenin signaling pathway during embryonic midgut development (p. 19, ll. 22-23).

Version 2:

Reviewer comments:

Reviewer #2

(Remarks to the Author)

The authors provided convincing evidence to address my concerns on nutrient digestion and absorption.

Reviewer #3

(Remarks to the Author)

In the revised manuscript, the authors added additional data comparing developing embryos and Squid mutant embryos. In addition, they described the limitations of this study to avoid overstatement of their findings. Therefore, the manuscript is now acceptable after the following corrections are made:

- In Figures 2D' and 2E', small scale bars overlap with the main scale bars. They should be removed. In addition, the scale bars in figure 2D" and 2E" are unclear. The color of the scale bars and "10 μ m" should be changed to improve visibility. An unnecessary scale bar is also shown on the left side of Figure 2E". This should be eliminated.

Responses to reviewers' comments

Dear reviewers,

We sincerely appreciate your thorough review and valuable feedback on our manuscript. As your concerns, there are several issues that need to be addressed. According to your insightful comments and suggestions, we have re-conducted and supplemented the relevant experiments, adjusted the presentation of certain figures, revised textual descriptions, and expanded the discussion of key results to improve its clarity and facilitate the understanding of the readers. Our detailed point-by-point responses are provided below. We hope that these revisions adequately address the reviewers' concerns and meet the journal's standards.

Reviewers' comments:

Reviewer #1 (Remarks to the Author):

Tong *et al.* present a study of the role of Squid in *Bombyx mori*, the silkworm. They find a role in midgut development during embryogenesis, which they characterize morphologically. They then show that Squid regulates the alternate splicing of Axin, potentially tying Wnt signaling to the morphological and genetic changes in midgut development seen in Squid mutants.

Unfortunately, although well carried out technically, Justification for the study and support for the broad conclusions was not strong. Squid is likely ubiquitously expressed in *Bombyx*. Why focus on midgut? Although it is expressed there, it is likely expressed elsewhere as well. Similarly, why use *Bombyx*? Is there something special about this insect that drives the study of Squid in this species? There are many reasons to study diverse animal models but is there a specific reason to study Squid in the midgut in this species? Is Squid expressed and/or active in the *Drosophila* midgut? Finally, as described below, the link between Axin alternative splicing and the defects seen in midgut morphology are correlative as presented here. Additional work would need to be done to test whether or not these change in Axin are causal.

A: Thank you for your carefully reviewing and constructive comments and suggestions. Since its establishment, our laboratory has focused on Lepidoptera physiology, primarily using the model insect *Bombyx mori* as our research subject. This manuscript builds upon our previous work (Deng *et al.*, 2012; Zhu *et al.*, 2021; Tong *et al.*, 2023). Here, we focused on the midgut because the *Squid* mutant line exhibited obvious abnormalities in midgut structure and function. Thus, the manuscript primarily explored Squid's role in the *B. mori* midgut. Notably, we also found Squid expressed in multiple tissues of *B. mori* embryos, and we are currently investigating its function in the epidermis (Tong *et al.*, 2023). Additionally, *Squid* is expressed in various *Drosophila* tissues, including the embryonic and larval midgut (<https://flybase.org/reports/FBgn0263396>).

As correctly noted, the current data demonstrate an association

between *Axin* alternative splicing and midgut morphological defects, but further experiments are indeed required to establish causality. To confirm this, it would be better to perform genetic rescue experiments by expressing wild-type *Axin* isoforms in mutant backgrounds to determine if splicing variants directly reverse the midgut phenotype and conduct tissue-specific knockdown/overexpression of specific *Axin* splice variants to validate their functional roles in midgut development. However, this is currently difficult to achieve in *B. mori* and we have included this limitation in the Discussion section of the revised manuscript. We acknowledge the critical importance of these mechanistic validations. To investigate the functional differences between *Axin* isoforms, we overexpressed Axin-L and Axin-S in *BmN* cells and analyzed their effects on β -catenin levels. Our results demonstrated that Axin-L overexpression significantly increased β -catenin levels, whereas Axin-S overexpression had no observable effect (Fig. 5G-5H of the revised manuscript), suggesting that Axin isoforms have different effect on β -catenin levels. This finding is similar to the previous reports showing that AXIN1 isoforms generated through exon 9 splicing exert opposing effects on β -catenin levels, thereby modulating distinct roles in hepatocellular carcinoma (HCC) cell migration and invasion (Zhang *et al.*, 2025). However, in contrast to the previous reports (Zhang *et al.*, 2025), our experiments in *BmN* cells showed that overexpression of Axin-L-EGFP increased β -catenin levels, while Axin-S-EGFP had no significant effect compared to the EGFP control. This discrepancy may reflect cell type-specific differences, as *BmN* cells are derived from ovary rather than embryonic tissue. Nevertheless, these *in vitro* results suggest that alternative splicing of *Axin* produces isoforms with distinct functional roles in modulating Wnt/ β -catenin signaling. Furthermore, we confirmed β -catenin levels in the embryonic midgut and observed a significant reduction in its level following Squid depletion (Fig. 5D-5F''). Taken together, our results indicate that *Axin* alternative splicing might influence embryonic midgut development by modulating the Wnt/ β -catenin signaling pathway. Since we examined the whole embryo rather than changes in *Axin* isoforms specifically within the embryonic midgut, we have revised the relevant conclusions in our manuscript to state that these results demonstrate that Squid regulates Wnt/ β -catenin signaling pathway likely through modulating the alternative splicing of *Axin* gene. Additionally, we also revised the related discussion.

Figure 5 (G) Effect of overexpression of Axin-L and Axin-S on the β -catenin content in *BmN* cells, respectively, detected by immunohistochemistry staining. Red fluorescent signals indicate β -catenin protein. Green fluorescence indicates overexpressed EGFP (b), Axin-L-EGFP (f) or Axin-S-EGFP (j) protein. Nuclei are marked by DAPI in cyan. The bars represent 50 μ m. (F) Quantified red fluorescence intensity of β -catenin per *BmN* cell after overexpression of EGFP, Axin-L-EGFP or Axin-S-EGFP protein, respectively. Different letters above the bars denote statistically significant differences analyzed by two-way ANOVA.

References:

- Deng HM, Zhang JL, Li Y, Zheng SC, Liu L, Huang LL, Xu WH, Palli SR, Feng QL. Homeodomain POU and Abd-A proteins regulate the transcription of pupal genes during metamorphosis of the silkworm, *Bombyx mori*. *Proc Natl Acad Sci USA.*, 2012; 109:12598-12603.
- Zhu Z, Tong C, Qiu B, Yang H, Xu J, Zheng S, Song Q, Feng Q, Deng H. 20E-mediated regulation of *BmKr-h1* by BmKRP promotes oocyte maturation. *BMC Biol.* 2021; 19(1):39.
- Tong CM, Zhang K, Rong ZX, Mo WY, Peng YL, Zheng SC, Feng QL, Deng HM. Alternative splicing of *POUM2* regulates embryonic cuticular formation and tanning in *Bombyx mori*. *Insect Sci.*, 2023; 30(5): 1267-1281.
- Zhang QQ, Miao YS, Hu JY, Liu RX, Hu YX, Wang F. The truncated AXIN1 isoform promotes hepatocellular carcinoma metastasis through SRSF9-mediated exon 9 skipping. *Mol Cell Biochem.*, 2025; (480):2247-2263.

Specific comments

Figure 1 - *B. mori* Squid CRISPR appears to have worked well and the large decrease in Squid protein is convincing. However, more information is needed about the process-number of embryos injected, number surviving to various stages, including adult, and number of mutants obtained. Most importantly, it seems as if just one single heterozygote was recovered, suggesting that all animals studied result from a single event in one animal. This is supported by sequence showing the same 2 bp deletion in all sequences. This may be problematic as it is essential to ensure that there is no other mutation in this line that could contribute to the mutant phenotype. Having multiple

mutant lines, with different mutations that resulted from different events, ideally from cleavage by different gRNAs would provide confidence that the Squid mutation is causal. If not that, at least 2-3 generations of backcrosses should be done. Ideally, rescue by wild type Squid would be definitive evidence, although transgenesis would be required and may be problematic in *Bombyx*.

A: We appreciate your constructive feedback. In the G0 generation, we injected 300 eggs, achieving a hatching rate of 17% (51 larvae). Ten individuals survived to adulthood, and only one was confirmed as a mutant (genotyped via sequencing). This founder mutant was outcrossed to wild-type, and subsequent progeny selfing yielded six distinct genotypes (Fig. R1). From these, we maintained the stable 2-bp deletion line using the following strategy: for homozygous mutants: heterozygous individuals were intercrossed, for population maintenance: heterozygotes were backcrossed to wild-type for $\geq 7-8$ generations to eliminate potential background mutations affecting the phenotype. We had added more details in the revised manuscript (Section Materials and Methods, Page 5-6). We note that *Squid* homozygous mutants are embryonic lethal in *B. mori*, but transgenic rescue remains technically challenging in this system.

ATGGAATGCAGAGAACGGTGGCGGTGATTGCGAAGATCATAACAGTGTGAGGCCCGCAGGACGC-GATGACGACAGGTAATAATTTCCCTTTTGTGAGTTT	Wild type
ATGGAATGCAGAGAACGGTGGCGGTGATTGCGAAGATCATAACAGTGTGAGGCCCGCAGGACGCTGA-GAACACGGGCTCTT--TTTTCCCTGTGTTGAGTGT	(-3)
ATGGAATGCAGAGAACGGTGGCGGTGATTGCGAAGATCATAACAGTGTGAGG-----G-C-----TGA-G---G--ACTGAATG--CCCATTGGTTGAGTTT	(-19)
ATGGAATGCAGAGAACGGTGGCGGTGATTGCGAAGATCATAACAGTGTGAGGCCCGCAGGACGATGA-GAACACGGGCACATA-TTT-CCCCTGTGTTGAGATTT	(-3)
ATGGAATGCAGAGAACGGTGGCGGTGATTGCGAAGATCATAACAGTGTGAGGCCCGCAGGACGC-G--GACACACGGGCTCTTTTTTTCTCTGTGTTGAGTTT	(-3)
ATGGAATGCAGAGAACGGTGGCGGTGATTGCGAAGATCATAACAGTGTGAGGCCCGCAGGACGCTGACAAC-ACGGGCAC--ATTTTTCTCTGTGTTGAGAGTT	(-3)
ATGGAATGCAGAGAACGGTGGCGGTGATTGCGAAGATCATAACAGTGTGAGGCCCGCAGGACGCTGA-GAACACGGGCTCTTTTTTTCCCT--GTGTTGAGTTT	(-3)

Figure R1. Screened for mutant genotypes of the G0 offspring

Pg. 9: the authors justify looking at midgut because they previous showed a role for Squid in the cuticle. I don't understand the logic.

A: Thanks for your valuable comment. Squid is highly expressed in the developing midgut epithelial cells and epidermis cells (Fig. 2A-2C), indicating its direct involvement in the development of both the midgut and epidermis. The function of Squid in the epidermis had been preliminarily discussed in our previous paper (Tong *et al.*, 2023). And the function and structure of the midgut were obviously abnormal in the *Squid* mutant line, thus, we mainly focused on the midgut in this manuscript. We have carefully revised the inappropriate descriptions in the revised manuscript (Section Results, Page 11).

References:

Tong CM, Zhang K, Rong ZX, Mo WY, Peng YL, Zheng SC, Feng QL, Deng HM. Alternative splicing of *POUM2* regulates embryonic cuticular formation and tanning in *Bombyx mori*. *Insect Sci.*, 2023; 30(5): 1267-1281.

Figure 2 – defects in the midgut seem convincing, although that is not the expertise of

this reviewer

A: Thank you for your recognition.

Figure 3: Why was RNA-seq carried out on whole embryos rather than dissected midguts? Also, what stage were these embryos? These choices need to be explained and justified since *Squid* is likely involved in many different processes. Bottom of page 10 states that the KEGG analysis points to the midgut. But many of the processes highlighted take place in many different cells and tissues of the animal. This conclusion needs to be critically analyzed and/or better explained.

A: Thanks for your valuable comments. In the early stages of this study, we conducted RNA-Seq analysis to investigate the potential genome-wide effects of *Squid* deletion in whole embryos, prior to discovering the midgut phenotype. Our rationale for selecting day 7 post-oviposition embryos for RNA-Seq based on the following observations: this represents the earliest developmental stage at which homozygous mutants can be reliably identified, as evidenced by the absence of head pigmentation (Fig. 1C-1D), while wild-type and heterozygous embryos display normal black head coloration. This time point allowed us to capture the primary transcriptional changes resulting from *Squid* knockout, before secondary effects accumulated. We have added the details in the revised manuscript (Section Materials and Methods, Page 5)

Lipids in eggs accumulated in oogenesis is the most important supply of energy for the developing embryo (Fruttero *et al.*, 2017). The midgut formed during embryonic development will encapsulate these lipids, and then digest the lipids and supply the nutrient and energy (Chapman, 2013). To investigate whether the abnormal development of midgut in *Squid*^{-/-} mutants suppresses the nutrition and energy supply, we detected the lipids in the midgut lumen with Nile Red staining. The results showed that there was no obvious accumulation of lipid droplets in the midgut lumen of the wild-type embryos, while a large number of lipid droplets (yellow arrow) were found in the midgut lumen of the *Squid*^{-/-} embryos (Fig. 3A a-f of the revised manuscript). Besides, the down-regulated DEGs in *Squid*^{-/-} embryos were enriched in the pathways associated to nutrient absorption and energy supply (Fig. 3B). Then the added RT-qPCR analysis revealed significant downregulation of all tested nutrient digestion- and absorption-related genes in the *Squid*^{-/-} embryos compared with the wild-type (Fig. 3C of the revised manuscript). Therefore, these results demonstrate that *Squid* deficiency severely disrupts midgut digestive function and nutrient absorption capacity. We have added the above results in the revised manuscript (Section Results, Fig. 3A and 3C, page 13-14).

Figure 4: (1) As for the RNA-seq, RIP-seq should have been carried out on dissected midguts at the stage of development when *Squid* would be expected to be active. The authors state that “embryos in organogenesis” were used. It is not clear what that means or what/why this stage was chosen. On the technical side, this approach seems to have been well executed and carefully analyzed. (2) The binding to coding sequences is unexpected. Were unspliced, full length RNAs recovered in the RIP-seq? Would binding to introns, splice borders and also 3’UTRs not be expected for a general RNA binding protein? (3) Genes differentially spliced are shown to fall into two major

pathways (Wnt and AGE-RAGE). How many of the 72 genes were in each of these pathways? Were Axin and Smad4 the only Wnt-pathway genes found? If not, why were these chosen for further analysis? Are they just good candidates or was there other evidence?

A: Thanks for your valuable suggestion. The organogenesis stage is 76 hours post oviposition, when the embryo begins to form organs, including the midgut. Our previous study showed that the *Squid* mRNA levels were high during organogenesis (Tong *et al.*, 2023). And we wanted to screen for genes that bind to Squid and regulate the formation of the midgut, therefore, the early stage of midgut development, organogenesis, was selected. In organogenesis stage, the midgut is not formed completely and difficult to dissect in *B. mori*, so we chose the whole embryo for RIP-seq.

The RNA precipitated by RIP-Seq is all the RNA that can bind to the target protein in the cell, mainly including mRNA, but also pre-mRNA, lncRNA, miRNA, snoRNA, etc. However, the library building method resulted in only long chain RNA being recovered, mainly including mRNA, pre-mRNA and lncRNA. Alternative splicing of individual pre-mRNAs is frequently controlled by *cis*-acting regulatory sequences and splicing factors that recognize and bind to these sites. These sites can be either intronic or exonic and can be positive (splicing enhancers) or negative (splicing silencers) (Lee and Rio, 2015). For example, hnRNP A1 binds to upstream of exon 3 in HIV *Tat* pre-mRNA and prevents binding of the U2 snRNP (Tange *et al.*, 2001). Tissue-specific splicing factors FOX1 and FOX2 inhibit the formation of the E' complex by binding to an intronic sequence to prevent SF1 from binding to the branch site of *CALCA* (calcitonin-related polypeptide- α) pre-mRNA (Zhou and Lou, 2008).

Four of the 72 genes were enriched in the Wnt signaling and AGE-RAGE signaling pathway. However, only *Axin* (BGIBMGA007135) and *Smad4* (BGIBMGA010110) genes were found in the Wnt signaling pathway, and *Plc-1* (BGIBMGA009604) and *Smad4* (BGIBMGA010110) were found in the AGE-RAGE signaling pathway.

References:

- Tong CM, Zhang K, Rong ZX, Mo WY, Peng YL, Zheng SC, Feng QL, Deng HM. Alternative splicing of *POUM2* regulates embryonic cuticular formation and tanning in *Bombyx mori*. *Insect Sci.*, 2023; 30(5): 1267-1281.
- Lee Y, Rio DC. Mechanisms and regulation of alternative pre-mRNA splicing. *Annu Rev Biochem.*, 2015; 84:291-323.
- Tange TO, Damgaard CK, Guth S, Valcarcel J, Kjems J. The hnRNP A1 protein regulates HIV-1 tat splicing via a novel intron silencer element. *EMBO J.*, 2001; 20:5748-5758.
- Zhou HL, Lou H. Repression of prespliceosome complex formation at two distinct steps by Fox-1/Fox-2 proteins. *Mol Cell Biol.*, 2008; 28:5507-5516.

Figure 5: The data on changes in Axin isoforms and b-catenin levels in the Squid mutant are convincing. However, it is unclear how this alternate splicing event would impact b-catenin levels. What is the function of the alternatively spliced region? Is Axin causal

here or is some other Wnt pathway gene involved? B-Catenin itself? Finally, the RT-PCR was done on whole embryos and should have been done on midgut. Thus, two aspects of conclusions such as this one on page 15 are not fully justified “Our study reveals that Squid is involved in the Wnt signaling pathway by regulating the alternative splicing of Axin during embryonic midgut development”- (1) there is no evidence that the alternative splicing of Axin causes the changes in b-catenin levels and (2) there is no evidence that the alternative splicing of Axin occurs in the midgut.

A: We sincerely appreciate the reviewer’s insightful questions regarding the mechanistic link between *Axin* alternative splicing and β -catenin regulation. We agree that our current data demonstrate a correlation rather than direct causation. As noted by the reviewer, the functional role of the alternatively spliced region in *Axin* remains to be determined. To investigate the functional differences between Axin isoforms, we overexpressed Axin-L and Axin-S in *BmN* cells and analyzed their effects on β -catenin expression. Our results demonstrated that Axin-L overexpression significantly increased β -catenin levels, whereas Axin-S overexpression had no observable effect (Fig. 5G-5H of the revised manuscript). This result is similar to the previous reports showing that AXIN1 isoforms generated through exon 9 splicing exert opposing effects on β -catenin expression, thereby modulating distinct roles in hepatocellular carcinoma (HCC) cell migration and invasion (Zhang *et al.*, 2025). However, in contrast to the previous reports (Zhang *et al.*, 2025), our experiments in *BmN* cells showed that overexpression of Axin-L-EGFP increased β -catenin levels, while Axin-S-EGFP had no significant effect compared to the EGFP control. This discrepancy may reflect cell type-specific differences, as *BmN* cells are derived from ovary rather than embryonic tissue. Nevertheless, these *in vitro* results suggest that alternative splicing of *Axin* produces isoforms with distinct functional roles in modulating Wnt/ β -catenin signaling. Why does Axin-L, which contains all structural domains, enhance β -catenin levels, whereas Axin-S, lacking a 46-amino acid segment that includes a low-complexity region (LCR), exerts no significant effect? Low complexity regions (LCRs) play an important role in a variety of important biological processes. Recent work has shown that in proteins with multiple LCRs, the contributions of individual LCRs on protein function can depend on their identities, and they can differentially contribute to the function of the protein, likely depending on their abilities to interact with different sequences (Lee *et al.*, 2022). Thus, it is rational to assume that the LCR may be dispensable for the function of the *Axin*, or alternatively, that its absence could confer distinct functional properties. Of course, the hypothesis that alternative splicing-mediated inclusion or exclusion of LCRs can contribute to the functional diversity of Axin proteins remains to be further determined.

Furthermore, we confirmed β -catenin expression in the embryonic midgut and observed a significant reduction in its level following Squid depletion (Fig. 5D-5F”). Therefore, our results indicate that *Axin* alternative splicing might influence embryonic midgut development by modulating the Wnt/ β -catenin signaling pathway. Since we examined the whole embryo rather than changes in *Axin* isoforms specifically within the embryonic midgut, we have revised the relevant conclusions in our manuscript to

state that these results demonstrate that Squid regulates Wnt/ β -catenin signaling pathway likely through modulating the alternative splicing of *Axin* gene. Additionally, we also revised the related discussion.

Figure 5 (G) Effect of overexpression of Axin-L and Axin-S on the β -catenin content in *BmN* cells, respectively, detected by immunohistochemistry staining. Red fluorescent signals indicate β -catenin protein. Green fluorescence indicates overexpressed EGFP (b), Axin-L-EGFP (f) or Axin-S-EGFP (j) protein. Nuclei are marked by DAPI in cyan. The bars represent 50 μ m. (F) Quantified red fluorescence intensity of β -catenin per *BmN* cell after overexpression of EGFP, Axin-L-EGFP or Axin-S-EGFP protein, respectively. Different letters above the bars denote statistically significant differences analyzed by two-way ANOVA.

We acknowledge that whole-embryo RT-PCR cannot definitively localize splicing events to the midgut. This technical limitation arises because: the embryonic midgut of *B. mori* at this stage is microscopic and cannot be cleanly dissected. The attempts to perform laser-capture microdissection were hindered by the chitinous eggshell. However, the results of immunohistochemistry data (Fig. 5E-5F'') showed that β -catenin accumulation was in the basal region of the midgut epithelium of the wild-type embryos on day 7 but not of the *Squid*^{-/-} embryos, confirming that Wnt/ β -catenin signaling exists in the embryonic midgut and *Squid* depletion inhibited Wnt/ β -catenin signaling in the embryonic midgut. We have added this limitation in the discussion section of the revised manuscript.

We fully agree that other factors (e.g., β -catenin stability regulators) may contribute. Therefore, we detected the expression change of the other key Wnt components (*APC* and *GSK3 β*) in the *Squid*^{-/-} embryos using RT-qPCR, and the results showed that the expression of *APC* had no obvious change while the expression of *GSK3 β* was slightly decreased in the *Squid*^{-/-} embryos (Fig. R2), suggesting that Squid might be also involved in the regulation of *GSK3 β* . Since Squid is the splicing regulatory factor and the analysis of RNA-seq and RIP-Seq identified *Axin* and *Smad4* as the direct target genes of Squid, we mainly focus on *Axin*.

Figure R2. RT-qPCR analysis of *BmAPC* (A) and *BmGSK-3β* (B) in wild-type and *Squid*^{-/-} embryos. The significance of the differences between the wild-type and *Squid*^{-/-} embryos was statistically analyzed using *t* tests at $p < 0.05$ (*).

References:

- Zhang QQ, Miao YS, Hu JY, Liu RX, Hu YX, Wang F. The truncated AXIN1 isoform promotes hepatocellular carcinoma metastasis through SRSF9-mediated exon 9 skipping. *Mol Cell Biochem.* 2025, 480(4):2247-2263.
- Lee B, Jaber-Lashkari N, Calo E. A unified view of low complexity regions (LCRs) across species. *Elife*, 2022; 13(11):e77058.

Finally, although embryonic lethality is clear and convincing, the midgut changes are not necessarily the cause of lethality, as claimed, for example, in the last sentence of the Discussion. Many other embryonic processes are likely implicated, as evidenced by the large number of change in the RNA-seq and RIP-seq experiments done on whole embryos.

A: Thank you for your insightful comment. We have revised the concluding statement of the Discussion to “Specifically, Squid regulates Wnt/ β -catenin signaling pathway likely through modulating the alternative splicing of *Axin* gene. The absence of Squid results in severe midgut developmental and functional impairments during embryogenesis, which is likely a major contributor to the embryonic lethality observed in *Squid* mutants.”.

Minor Points

There are small grammar mistakes in various places in the manuscript (eg, noun-verb agreement). These mistakes are quite minor and do not hamper understanding.

A: Thanks. These mistakes have been revised in the manuscript.

Yellow dots and many of the letters are hard to see in Figure 2

A: Thanks for your suggestion! These have been changed in the revised Fig. 2.

Figure 5E – b-catenin cannot be seen in merge (E'')

A: Thanks for your comment! We have re-merged the image (E) and (E') as (E'') and the image (F) and (F') as (F''), shown as Fig. 5 of the revised manuscript.

Response to specific questions

Does the manuscript have technical or conceptual flaws that should prohibit its publication?

There are no technical flaws. The experiments are well-designed and carefully carried out from a technical point of view.

Are the conclusions original?

The conclusions are original but not strongly supported by the data.

Do you feel that the results presented are of immediate relevance for people in your own discipline or for a broader audience?

I feel this result is incremental and the authors have not justified or explained the broad interest.

Reviewer #2 (Remarks to the Author):

In this study the authors investigated the role of Squid, a member of hnRNPs family, in midgut development in the silkworm, *Bombyx mori*. After demonstrating that Squid knockout impairs midgut formation during embryogenesis, they identified target genes and demonstrated that one of them (Axis) undergoes alternative splicing mediated by Squid.

As indicated in the article, although insect midgut has been widely analyzed from different points of view, the molecular mechanisms regulating its development are scarcely known. Thus, novel information on this topic is welcome, especially if it has been obtained in a model insect as the silkworm. However, I think that the manuscript presents many flaws and, in particular, five key aspects need to be carefully addressed:

1. The quality of some images is poor and it's thus difficult to finely evaluate the effects of *Squid* knockout on the midgut. In particular, some images cannot be compared (wt VS *Squid*^{-/-}) and additional pictures must be provided to substantiate the authors' conclusions as detailed below.

A: Thank you for your carefully reviewing and constructive comments and suggestions. Some of the vague and unclear images may be due to them being exported directly from PowerPoint after layout. We have uploaded the high-resolution images separately in the submission system.

2. Although this represents a key aspect of the study, the authors did not provide any

direct experimental evidence demonstrating the impairment of nutrient digestion and absorption after *Squid* knockout, except showing a variation in the amount of lipid droplets in the midgut lumen. Moreover, also intestinal stem cells seem to be affected by impairing *Squid* signaling but, once again, no specific experiments have been performed to this purpose. This lack of data significantly affects the characterization of the role of *Squid* in this setting and, consequently, reduces the impact of the study.

A: Thank you for your constructive comments! We sincerely appreciate the reviewer's valuable feedback. In response to these insightful comments, we have incorporated new experimental evidence in the revised manuscript that systematically demonstrates impaired nutrient digestion and absorption following *Squid* knockout. Lipids in eggs accumulated in oogenesis is the most important supply of energy for the developing embryo (Fruttero *et al.*, 2017). The midgut formed during embryonic development will encapsulate these lipids, and then digest the lipids and supply the nutrient and energy (Chapman, 2013). To investigate whether the abnormal development of midgut in *Squid*^{-/-} mutants suppresses the nutrition and energy supply, we detected the lipids in the midgut lumen with Nile Red staining. The results showed the striking differences between genotypes: while wild-type embryos showed minimal lipid droplet accumulation in the midgut lumen (Fig. 3A a-c of the revised manuscript), *Squid*^{-/-} mutants exhibited substantial lipid retention (indicated by yellow arrows, Fig. 3A d-f of the revised manuscript). Besides, the downregulated DEGs in *Squid*^{-/-} embryos were enriched in the pathways associated to nutrient absorption and energy supply (Fig. 3B). Also, we used RT-qPCR to detect the expression levels of genes related to nutrient digestion and absorption in the 7-day-old wild-type and *Squid*^{-/-} embryos. The results showed that all detected nutrient digestion and absorption related genes were significantly downregulated in the *Squid*^{-/-} embryos compared with the wild-type (Fig. 3C of the revised manuscript). Combined with the phenotypes observed by electron microscopy, we demonstrated that nutrient digestion and absorption of *Squid*^{-/-} embryos were blocked at the morphological, histochemical and genetic levels. We have added the above results in the revised manuscript (Section Results, Fig. 3A and 3C, page 13-14).

Figure 3. Knockout of *Squid* inhibits nutrition and energy metabolism. (A) Nile red staining of the midgut in wild-type and *Squid*^{-/-} embryos on day 7 after oviposition. The yellow arrow represents lipid droplets. ME, midgut epithelium. L, lumen. Nuclei are marked by DAPI in cyan. The bars represent 50 μm. (B) KEGG enrichment pathways for genes that are down-regulated in *Squid*^{-/-} embryos. (C) qRT-PCR validation of nutrient digestion and absorption-related genes in wild-type and *Squid*^{-/-} embryos on day 7 after oviposition. *Atp5me*, *H*⁺ transporting ATP synthase subunit *e*. *ATP-PFK*, ATP-dependent 6-phosphofructokinase. *HGD*, Homogentisate 1,2-dioxygenase. *HAGH*, hydroxyacylglutathione hydrolase. *IDH*, Isocitrate dehydrogenase. *PDHB*, pyruvate dehydrogenase E1 component beta subunit. The significance of the differences between the WT and MT lines was statistically analyzed using *t* tests at *p* < 0.01 (**) and *p* < 0.0001 (****).

References:

- Fruttero LL, Leyria J, Canavoso LE. Lipids in Insect Oocytes: From the Storage Pathways to Their Multiple Functions. *Results Probl Cell Differ*. 2017; 63:403-434.
- Chapman RF, The insects: structure and function. Simpson SJ, Douglas AE (eds) Cambridge University Press, Cambridge, 2013.

The primary function of intestinal stem cells (ISCs) are proliferation and differentiation. To detect whether *Squid* knockout affects the proliferation of intestinal stem cells, we performed immunohistochemistry using an antibody against phospho-histone H3 (P-H3), a conserved proliferation marker, on 7-day-old wild-type and *Squid*^{-/-} embryos. The results showed that P-H3 signals were undetectable in both genotypes (Fig. S4a-f), suggesting that the proliferation of ISCs was largely complete by this stage. To validate antibody efficacy, we conducted parallel experiments on fifth-instar larval

wing discs, which showed robust P-H3 signals (indicated by white arrows, Fig. S4g-i), confirming the antibody's effectiveness in *B. mori*. Besides, the results of TEM showed that the differentiation of wild-type midgut epithelium cells was normal and completely, and the morphology of ISCs, CCs and GCs were clearly and can be obviously distinguished (Fig. 2D’'). However, the shapes of ISCs, CCs and GCs in *Squid*^{-/-} midgut were all altered, with only GCs exhibiting smaller cavities being recognizable; CCs and ISCs could not be distinctly identified (Fig. 2E’'). These results suggest that epithelial cells differentiation is abnormal in *Squid*^{-/-} midgut. Additionally, further immunohistochemical analysis showed that Squid proteins were expressed in the midgut epithelial cells of 7-day-old wild-type embryos (Fig. S4a-c), whereas it is disappeared in the midgut epithelial cells of 7-day-old *Squid*^{-/-} embryos (Fig. S4d-f). Based on these results, we speculate that Squid does not regulate ISCs proliferation in the post embryonic development, but may participate in the differentiation of embryonic ISCs. We have added the related results in the revised manuscript (Section Results, Fig. S4; Page 12-13).

Figure S4. Detection of proliferation signals (A) and expression level of Squid proteins (B) in wild-type and *Squid*^{-/-} embryonic midgut at day 7 after oviposition. The white arrow represents proliferation signals. The orange arrow represents proliferation signals. ME, midgut epithelium. L, lumen. The bars in (A) and in (B) represent 50 μm and 10 μm, respectively.

3. Along the manuscript, the authors draw a parallelism between i) the silkworm and *Drosophila*, and ii) the silkworm and mammals. It is not completely clear to me if this is a pioneering study and these regulatory mechanisms have been here characterized for the first time in the silkworm (this would provide a significant novelty to the current study) or the present study aims at verifying the existence of homologous mechanisms in this insect starting from previous evidence obtained in the fruitfly. In relation to the second aspect indicated above, I did not understand if this study aims at paving the way to future studies in mammalian models. In this case, please consider that the insect midgut and the human digestive system are significantly different.

A: We sincerely appreciate the reviewer’s insightful feedback. The regulatory mechanism identified in this study represents the first reported instance of such regulation in silkworm. To our knowledge, no homologous mechanisms have been

described in other species to date. This study mainly aimed to investigate Squid's role in regulating the mechanism on embryonic midgut development in silkworm. We are sorry that the description in the manuscript did not sufficiently emphasize this focus. In the revised manuscript, we have highlighted the originality of this study and clarify the scope by refining ambiguous descriptions throughout the text.

4. Materials and methods. Some additional information is necessary to reproduce the experiments.

A: Thanks for your suggestion. We have carefully reviewed the materials and methods and added related information in the revised manuscript.

5. Discussion must be significantly improved. Currently this section of the manuscript contains a summary of the main results integrated with some papers taken from the literature. However, the cited literature is sometimes inconsistent, not appropriately linked and does not contribute to develop a critical Discussion. Please note that it is not necessary to mention figures in this section.

A: Thanks for your valuable comments. In the revised manuscript, we have made major revisions to the Discussion based on the reviewer's constructive suggestion. We have carefully checked and revised the references and removed the figures labels.

- P3 L10. This reference does not directly support the point being made.

A: Thank you for your comments. We have corrected the reference in the revised manuscript.

- Paragraph 2.2. How did they perform the screening of insects and the characterization of the obtained lines?

A: Thanks for your valuable comments. To screen for mutant individuals in the G0 generation without compromising reproduction or development, genomic DNA was extracted from larval exuviae collected prior to pupation. The Squid locus was subsequently amplified and sequenced to determine the genotype. Consistent with previous observations, heterozygous mutants (*Squid*^{+/-}) exhibited normal growth, development, and fertility; therefore, exuviae-based genotyping was routinely used to identify these carriers. In contrast, homozygous mutants (*Squid*^{-/-}) displayed markedly delayed embryonic development at 7-8 days post-oviposition (Fig. 1C-D). Whereas wild-type and heterozygous embryos exhibited visible cephalic pigmentation by day 7, homozygous embryos lacked this characteristic melanization (Fig. 1C-D). Consequently, eggs with overt developmental delay were isolated, and the *Squid* gene was amplified and sequenced to confirm the homozygous mutant genotype. These methodological details have been incorporated into the revised manuscript.

- P5 L23-24. What do they mean with "sections were affixed to the slides"?

A: Thanks a lot. It means the thin slices of embryos cut by ultramicrotome (EM UC7, Leica, Wetzlar, Germany) were attached to the slides. We have carefully revised the description (Section Materials and Methods, Page 7).

- Paragraph 2.3. Why did they use proteinase K on tissue sections? Moreover, information on the two antibodies (source, concentration of use, etc.) must be given.

A: Thank you for your comment. Antigen retrieval was performed by incubating deparaffinized tissue sections with proteinase K to partially digest formalin-induced cross-links and thereby expose masked epitopes. This enzymatic pretreatment significantly increases the accessibility of target antigens, resulting in enhanced sensitivity and specificity of subsequent immunohistochemical staining. Detailed information on the two primary antibodies has now been included in the revised manuscript (Section Materials and Methods, Page 6-7).

- P6 L8. How were proteins isolated from the tissue?

A: Thanks a lot. Wild-type and *Squid*^{-/-} eggs were homogenized in ice-cold RIPA lysis buffer (Beyotime, China) and clarified by centrifugation at 12,000 × g, 4 °C for 20 min. The resulting supernatants were passed through a 0.45 μm membrane filter and stored at -80°C until analysis. Total protein concentrations were determined using the Bradford assay (Bradford, 1976). These details have been incorporated into the revised manuscript (Section Materials and Methods, Page 6).

- P7 L12. Please provide information on the lysis buffer.

A: Thanks. The information of lysis buffer has been added in the revised manuscript (Section Materials and Methods, Page 8).

- P8 L3. Is RP49 adequate as housekeeping gene in this setting? Is its expression stable along development?

A: Thanks for your valuable comment. A previous study systematically evaluated the expression stability of multiple housekeeping genes in four lepidopteran species using geNorm, NormFinder, stability index, and ΔCt analyses. The results indicated that *RP49* and *GAPDH* exhibited the highest stability across different developmental stages in *B. mori* and *Spodoptera exigua*, respectively (Teng *et al.*, 2012).

References:

Teng X, Zhang Z, He G, Yang L, Li F. Validation of reference genes for quantitative expression analysis by real-time RT-PCR in four lepidopteran insects. *J Insect Sci.*, 2012; 12:60.

- P8 L14. Do they mean in *Bombyx mori*?

A: Thanks. Yes, it refers to *Bombyx mori*.

- P8 L26. Please note that in Fig. S2C the homozygote is reported.

A: Thank you for your reminder. We have corrected the Fig. S2B and Fig. S2C in the revised version.

- P9 L15 (and Figs. 2A-C). It is difficult to interpretate these images and the insect anatomy is quite strange. Why is the midgut so close to epidermis? Moreover, I do not see the fat body.

A: Thanks for your valuable comment. Lepidopteran larvae possess a tubular body plan in which an elongate midgut occupies the majority of the haemocoel and lies immediately adjacent to the epidermis. In the immunohistochemical analyses shown in Fig. 2A-C, we exploited this anatomical arrangement to localize Squid protein. The midgut's prominent lumen and the epidermis's position as the outermost cell layer provide unambiguous landmarks, whereas the fat body lacks distinctive morphological features under the conditions employed and therefore could not be reliably identified.

- P9 L16-19. The correlation of this previous evidence taken from the literature with the current paragraph is not clear.

A: Thanks so much. We have carefully revised the inappropriate descriptions in the revised manuscript (Section Results, Page 11).

- P9 L19 to P10 L3 (and Figs. 2D-E). How can the authors be sure that these are lipids? Did they perform any histochemical staining or biochemical analysis? Moreover, both figures are difficult to be interpreted (see lumen, microvilli, spherites, arrows, etc.). Finally, variation in stem cells must be clearly visible and specific (proliferation) assays should be used to this purpose.

A: Thanks for your valuable suggestion. In transmission electron micrographs, lipid droplets exhibit a distinctly rounded profile and high electron density, consistent with previous reports (Melo *et al.*, 2013; Sugiyama *et al.*, 2019). The droplets identified in the present study display identical morphological features (Fig. 2E''). To corroborate these ultrastructural observations, we performed Nile red staining, as illustrated in Fig. 3A of the revised manuscript.

Figure 3A. Nile red staining of the midgut in wild-type and *Squid*^{-/-} embryos on day 7 after oviposition. The yellow arrow represents lipid droplets. ME, midgut epithelium. L, lumen. Nuclei are marked by DAPI in cyan. The bars represent 50 μm.

The results and statistical analyses of the stem-cell proliferation assay are presented in Fig. S4 of the revised manuscript.

Figure S4. Detection of proliferation signals (A) and expression level of *Squid* proteins (B) in wild-type and *Squid*^{-/-} embryonic midgut at day 7 after oviposition. The white arrow represents proliferation signals. The orange arrow represents proliferation signals. ME, midgut epithelium. L, lumen. The bars in (A) and in (B) represent 50 μm and 10 μm, respectively.

References:

Melo RC, Paganoti GF, Dvorak AM, Weller PF. The internal architecture of leukocyte lipid body organelles captured by three-dimensional electron microscopy tomography. *PLoS One*. 2013; 8(3):e59578.

Sugiyama M, Shindo D, Kanada N, Ohzeki T, Yoshioka K, Funaba M, Hashimoto O. Inducible brown/beige adipocytes in retro-orbital adipose tissues. *Exp Eye Res.*, 2019; 184:8-14.

- P9 L24-25. Please consider that the midgut epithelium also contains endocrine cells.

A: Thanks a lot. It is well established that the midgut epithelium of *Drosophila* contains

endocrine cells, but it is not clear whether endocrine cells exist in the midgut epithelium of lepidopteran insects.

- P10 L12-14. Although the authors here declare that they performed transcriptomic analysis to demonstrate that Squid depletion impairs nutrient absorption, they do not provide any evidence showing that the expression of transcripts involved in nutrient digestion and absorption are modified in this organ. Moreover, they conclude that their data indicate that the depletion of Squid leads to abnormal embryonic midgut development, ultimately inhibiting overall embryonic development (P10 L23-25). I'm quite confused about this final statement.

A: Thanks for your valuable comments. In the revised manuscript, we provide new evidence that the expression of transcripts involved in nutrient digestion and absorption are significantly decreased in the 7-day-old *Squid*^{-/-} embryos, compared to the wild-type, being shown as Fig. 3C of the revised manuscript. Besides, we have revised this incorrect description in the revised manuscript.

Figure 3C. qRT-PCR validation of nutrient digestion and absorption-related genes in wild-type and *Squid*^{-/-} embryos on day 7 after oviposition. *Atp5me*, H⁺ transporting ATP synthase subunit *e*. *ATP-PFK*, ATP-dependent 6-phosphofructokinase. *HGD*, Homogentisate 1,2-dioxygenase. *HAGH*, hydroxyacylglutathione hydrolase. *IDH*, Isocitrate dehydrogenase. *PDHB*, pyruvate dehydrogenase E1 component beta subunit. The significance of the differences between the WT and MT lines was statistically analyzed using *t* tests at $p < 0.01$ (**) and $p < 0.0001$ (****).

- P13 L1-2 (and Figs. 5E-F). The fluorescent signal shown in these figures does not allow to assess the precise localization of B-catenin in midgut samples. Moreover, did the authors hypothesize any cytoplasm-to-nucleus translocation of B-catenin as indicated in the Discussion? Finally, the figures needs labels and orientation should be reconsidered.

A: Thanks for your valuable suggestion. The images presented in Fig. 5E-F” do not

provide sufficient resolution to definitively localize β -catenin to the midgut, so our description of which “the basal region of the midgut epithelium” is a relatively vague conclusion. As shown in the Fig. 5E-E”, β -catenin were mainly expressed in the cytoplasm at midgut. However, because cytoplasmic-to-nuclear translocation of β -catenin is not the focus of the present study, it was not discussed in detail. All micrographs have been relabeled and reoriented for clarity in the revised figures.

- Figure 6. Please indicate stem cells as “ISCs”.

A: Thanks a lot. “ISCs indicate intestinal stem cells” has been added in the legend of Fig. 6.

- Figures. Scale bars must be revised (see font dimension and orientation).

A: Thanks. It has been changed in the revised figures.

- The paper needs a language revision by a mother tongue or a professional editing service. There is a misuse of some terms and the paper contains many typos.

A: Thanks for your valuable suggestion. We have found senior scientist for language revision before finalization.

Reviewer #3 (Remarks to the Author):

In the current manuscript, Tong et al. examine molecular function of a heterogeneous nuclear ribonucleoprotein, Squid, in embryonic development in the silkworm, *Bombyx mori*. The authors previously demonstrated that a target gene of Squid is responsible for cuticular development during embryogenesis (Tong et al., 2023). In this manuscript, the authors generated a Squid gene knockout (KO) silkworm by CRISPR-based mutagenesis. The Squid KO silkworms died during embryogenesis with defects in midgut development. In addition, the authors conducted transcriptome analysis and RNA-immunoprecipitation sequencing analysis to determine target genes of Squid that controls embryogenesis. The authors identified that Squid regulates alternative splicing of Axin, a Wnt signaling regulator.

The authors performed multiple analyses to demonstrate molecular function of Squid. In particular, they determined the binding motif of Squid and the target genes of Squid. However, the current manuscript contains critical errors that makes it impossible to interpret the data properly.

Major points:

1) The major concern is the lack of evidence that Squid within the midgut regulates the midgut development during embryogenesis. As shown in Figure 1C and D, the embryonic development of the Squid KO silkworm is already delayed or arrested before day-7 after egg laying. The authors then compare the midgut development of 7-day-old

embryos between the wild-type and Squid KO silkworms. Their developmental stages are completely different at that time. Therefore, it is likely that the midgut of the Squid KO silkworm is not disordered as the authors describe (p.9, l. 21) but rather underdeveloped. Considering the high expression of Squid during early embryogenesis, as shown by the authors in the previous study (Tong et al., 2023), Squid is likely required for the early embryogenesis. Moreover, it is unclear whether Squid is specifically expressed in the midgut. The authors emphasize the expression of Squid in the midgut and epidermal cells (p. 9, ll. 14-15). However, Figure 2A-C shows that Squid is detected in most cells. The authors need to provide evidence showing that Squid regulates midgut development within the midgut cells.

A: Thank you for your carefully reviewing and constructive comments and suggestions. As shown in Figure 1C and D, the development of *Squid*^{-/-} embryos were delayed compared to the wild-type embryos at 7-9 days after egg laying. Therefore, it is reasonable to think that the midgut in the *Squid*^{-/-} embryos was underdeveloped. However, the subsequent immunohistochemical staining on the 7-day-old wild-type and *Squid*^{-/-} embryos using a conserved proliferation signal marker, phospho-Histone H3 (P-H3) antibody, showed that no proliferation signal was detected in the midgut of either wild-type or *Squid*^{-/-} embryos (Fig. S4Aa-f), indicating that midgut morphogenesis is complete in both genotypes at this stage. The validity of the antibody was confirmed by strong P-H3 labelling in fifth-instar larval wing discs processed in parallel (Fig. S4Ag-i), proving that P-H3 antibody could be effectively applied in silkworms. Furthermore, the results of TEM showed fully differentiated midgut epithelium in wild-type embryos, with clearly distinguishable intestinal stem cells (ISCs), columnar cells (CCs) and goblet cells (GCs) (Fig. 2D''). In contrast, all three cell types were morphologically altered in *Squid*^{-/-} embryos; only GCs—recognizable by their reduced cavities—could be identified, whereas ISCs and CCs were indistinct (Fig. 2E''). These data indicate defective epithelial differentiation in the absence of Squid. Additionally, immunohistochemistry confirmed robust Squid expression in the midgut epithelium of 7-day wild-type embryos, whereas Squid was undetectable in *Squid*^{-/-} embryos (Fig. S4Ba-f). The above results are presented in Fig. S4 of the revised manuscript. Collectively, these results support a role for Squid in the differentiation of embryonic ISCs during post-embryonic development. We note that Squid is expressed broadly and likely fulfils additional functions—for example, in epidermal morphogenesis (Tong *et al.*, 2023). Further dissection of ISC-specific roles is currently hindered by the limited availability of silkworm-specific antibodies and transgenic tools.

Figure S4. Detection of proliferation signals (A) and expression level of Squid proteins (B) in wild-type and *Squid*^{-/-} embryonic midgut at day 7 after oviposition. The white arrow represents proliferation signals. The orange arrow represents proliferation signals. ME, midgut epithelium. L, lumen. The bars in (A) and in (B) represent 50 μm and 10 μm, respectively.

2) There are serious errors in the analysis and conclusions of β -catenin. Considering the function of Axin as a negative regulator of β -catenin, Axin has weak activity in the wild type at Day-7, suggesting that Axin-1 does not function as a Wnt signaling regulator, but Axin-2 does. This is unlikely, as Axin-1 contains functional domains (Figure 5A). If this interpretation is correct, the amount and localization of β -catenin is not regulated by Axin, but rather controlled by other factors such as the Wnt ligand. Moreover, as described above, developmental stages of the wild-type and *Squid* KO silkworms are different. Wnt signaling fluctuates along with embryonic development (Mundaca-Escobar et al., Front. Cell Dev. Biol., 2022). Therefore, the authors should compare these at the same developmental stage.

A: Thanks for your valuable comments. It has been reported that AXIN1 isoforms resulting from AXIN1 exon 9 splicing, played distinct roles in the migration and invasion of HCC cells, namely, AXIN1-L decreased HCC cells migratory, invasive and proliferative ability, while AXIN1-S enhanced it. In details, Overexpression of AXIN1-L promoted the phosphorylation of β -catenin in both HCCLM3 and huh-7 cells, whereas overexpression of AXIN1-S inhibited the expression of p- β -catenin. Similarly, the knockdown of AXIN1-L inhibited the expression of p- β -catenin, whereas knockdown of AXIN1-S had the opposite effect (Zhang *et al.*, 2025). This provides the evidence that the alternative splicing of Axin-1 causes the changes in β -catenin levels. To investigate the functional differences between Axin isoforms, we overexpressed Axin-L and Axin-S in *BmN* cells and analyzed their effects on β -catenin levels. Our results demonstrated that Axin-L overexpression significantly increased β -catenin levels, whereas Axin-S overexpression had no observable effect (Fig. 5G-5H of the revised manuscript). This finding is similar to the previous report (Zhang *et al.*, 2025). However, our experiments in *BmN* cells showed that overexpression of Axin-L-EGFP increased β -catenin levels, while Axin-S-EGFP had no significant effect compared to the EGFP control. This discrepancy may reflect cell type-specific differences, as *BmN*

cells are derived from ovary rather than embryonic tissue. Nevertheless, these *in vitro* results suggest that alternative splicing of *Axin* produces isoforms with distinct functional roles in modulating Wnt/ β -catenin signaling. Future *in vivo* studies in genetically amenable systems, such as *Drosophila*, will be essential to validate this model. Additionally, why does Axin-L, which contains all structural domains, enhance β -catenin levels, whereas Axin-S, lacking a 46-amino acid segment that includes a low-complexity region (LCR), exerts no significant effect? LCRs are known to play crucial roles in diverse biological processes. Recent studies indicate that in proteins harboring multiple LCRs, the functional contribution of each LCR depends on its specific identity and its capacity to engage with distinct molecular sequences, thereby differentially influencing protein activity (Lee *et al.*, 2022). It is therefore plausible that the LCR missing in Axin-S may be nonessential for Axin function, or alternatively, that its absence endows the protein with altered functional characteristics. Of course, the hypothesis that alternative splicing-mediated inclusion or exclusion of LCRs can contribute to the functional diversity of Axin proteins remains to be further determined. Furthermore, we confirmed β -catenin expression in the embryonic midgut and observed a significant reduction in its level following Squid depletion (Fig. 5D-5F’). Therefore, our results indicate that *Axin* alternative splicing might influence embryonic midgut development by modulating the Wnt/ β -catenin signaling pathway. Since we examined the whole embryo rather than changes in *Axin* isoforms specifically within the embryonic midgut, we have revised the relevant conclusions in our manuscript to state that these results demonstrate that Squid regulates Wnt/ β -catenin signaling pathway likely through modulating the alternative splicing of *Axin* gene. Additionally, we also revised the related discussion.

Figure 5 (G) Effect of overexpression of Axin-L and Axin-S on the β -catenin content in *BmN* cells, respectively, detected by immunohistochemistry staining. Red fluorescent signals indicate β -catenin protein. Green fluorescence indicates overexpressed EGFP (b), Axin-L-EGFP (f) or Axin-S-EGFP (j) protein. Nuclei are marked by DAPI in cyan. The bars represent 50 μ m. (F) Quantified red fluorescence intensity of β -catenin per *BmN* cell after overexpression of EGFP, Axin-L-EGFP or Axin-S-EGFP protein, respectively. Different letters above the bars denote statistically significant differences analyzed by two-way ANOVA.

The immunohistochemical staining with an antibody against phospho-Histone H3 (P-H3), a conserved marker of proliferation, revealed no detectable proliferative signal in the midgut of either wild-type or *Squid*^{-/-} 7-day-old embryos (Fig. S4Aa-f), indicating that midgut morphogenesis is complete in both genotypes by this stage. Accordingly, we compared the expression of β -catenin protein in the midgut epithelium of 7-day-old wild-type and *Squid*^{-/-} embryos (Fig. 5E-F”), a time point at which midgut development is concluded in both backgrounds. Although it would be preferable to directly compare the expression of the *Axin* isoforms within midgut cells, this analysis is currently limited by the lack of silkworm-specific antibodies and transgenic tools. Future *in vivo* studies in *Drosophila*, the genetically amenable systems, will be essential to validate this model.

References:

- Zhang QQ, Miao YS, Hu JY, Liu RX, Hu YX, Wang F. The truncated AXIN1 isoform promotes hepatocellular carcinoma metastasis through SRSF9-mediated exon 9 skipping. *Mol Cell Biochem.* 2025, 480(4):2247-2263.
- Lee B, Jaberi-Lashkari N, Calo E. A unified view of low complexity regions (LCRs) across species. *Elife.* 2022, 13(11):e77058.

3) The source information for the anti-Squid polyclonal antibody is missing in the manuscript. The authors need to cite relevant articles if the antibody has already been used in previous studies. Otherwise, the authors need to indicate how to obtain the antibody. It is unclear whether the antibody specifically binds to Squid. Although the authors show that the Squid KO silkworm does not have the target protein (Figure 1F), they do not show the molecular size of bands or the immunocross-reactivity of the anti-Squid antibody. The authors should show the entire blot with molecular markers, as well as a control lane with the second-antibody only, if this is the first study to use this antibody. The authors should not use this antibody for immunohistochemistry (Figure 2A-C), if it cross-reacts with other target proteins. It is also confusing that there are two bands in Figure 1F. Although the authors mention that these are Squid-1 and Squid-2 (p. 9, ll. 6-7), they did not explain about Squid-2.

A: Thanks for your valuable comments. The anti-Squid polyclonal antibody was generated by ourselves, and the complete procedure has been added to the Materials and Methods section of the revised manuscript. Briefly, the full-length Squid ORF was cloned into pET-32a to yield an N-terminal His-tagged construct (Squid-pET-32a). Recombinant Squid-His (theoretical mass 51.28 kDa) was expressed in *Escherichia coli* BL21 (DE3) and purified by Ni²⁺-chelating affinity chromatography using a His-Bind® kit (Novagen, USA). After SDS-PAGE verification (Figure. S1A), the purified protein was emulsified 1:1 (v/v) with Freund's complete adjuvant and used to immunize New Zealand White rabbits subcutaneously (four boosts at 2-week intervals). Immune serum was collected 7 days after the final boost and stored at -80 °C. Specificity was confirmed by Western blot. The anti-Squid serum, but not pre-immune serum, detected purified Squid-His (Fig. S1B-C). The above results have been represented as Figure S1 of the revised manuscript.

In embryonic lysates, the antibody recognized two bands of ~31.68 kDa (Squid-1) and ~33.44 kDa (Squid-2; 48 bp longer isoform) in wild-type samples, whereas no corresponding bands were observed in *Squid*^{-/-} embryos (Fig. 1F), demonstrating high specificity of the prepared antibody. The uncropped and unedited blot images have been uploaded as Supplementary Figures in the Supplementary material.

Figure S1. Specificity analysis of the anti-Squid polyclonal antibody. (A) SDS-PAGE detection of purified recombinant protein Squid-His. Western blot analysis of the specificity of pre-immunization serum (B) and anti-Squid polyclonal antibody (C). M, Standard protein molecular weight.

4) The authors describe accumulation of lipid droplets in the midgut lumen in the Squid KO silkworm (Figure 2E). To the best of my knowledge, however, lipids cannot be stained by HE staining.

A: Thanks for your insightful comment. HE staining is not typically effective for visualizing lipid droplets, you're absolutely right to question this observation. Transmission electron microscopy revealed that lipid droplets in the midgut lumen of *Squid*^{-/-} embryos exhibit the characteristic round morphology and high electron density reported previously (Melo *et al.*, 2013; Sugiyama *et al.*, 2019) (Fig. 2E''). These ultrastructural findings were corroborated by Nile red staining, which confirmed the abundant presence of neutral lipids in the same region (Figure 3A of the revised manuscript). Although H&E-stained sections contained numerous clear, round vacuoles of comparable size (Fig. 2E'), we now conservatively describe these structures as "lipid droplet-like vacuoles" because H&E alone does not permit definitive lipid identification.

Figure 3A. Nile red staining of the midgut in wild-type and *Squid*^{-/-} embryos on day 7 after oviposition. The yellow arrow represents lipid droplets. ME, midgut epithelium. L, lumen. Nuclei are marked by DAPI in cyan. The bars represent 50 μm.

Other points should be corrected for the future submission:

- The sizes of the images in Figure 2D' and Figure 2E' are different. Figure 2E' may be twice as large as Figure 2D'. Also, horizontal scale bars should be added to all images.

A: We thank the reviewer for this careful observation. The scale bar in Fig. 2E' was incorrectly sized relative to Fig. 2D'; this has been corrected, and horizontal scale bars have been added to all micrographs in the revised figures.

- For immunohistochemistry, authors treated samples with Proteinase K (p. 6, l. 1). This treatment digests target proteins.

A: Thanks very much. Antigen retrieval was performed by incubating deparaffinized sections with proteinase K (10 min, room temperature). This brief digestion selectively cleaves formalin-induced protein cross-links, unmasking otherwise inaccessible epitopes without compromising target antigens, thereby enhancing the sensitivity and specificity of immunohistochemical detection. The 10 min treatment duration has been added to the Materials and Methods section of the revised manuscript.

- The authors used 30 μg of proteins for Western blotting (p. 6, l. 8). The methods for protein extraction and quantification should be provided.

A: Thanks a lot. The methods for protein extraction and quantification have added to the revised manuscript.

- The authors use the TopHat software for RNA sequencing. The TopHat developers recommend to use Hisat2 instead of TopHat (<https://ccb.jhu.edu/software/tophat/index.shtml>), as of low accuracy of TopHat.

A: Thank you for your reminder, we will use Hisat2 in the next experiment.

- Add reference articles of the RIP-seq software if any.

A: Thanks a lot. The references have added to the revised manuscript.

- The authors analyzed differentially binding peaks by the "python script" (p. 7, ll. 21-22). The detail should be provided. If it is on a public data base, that should be cited. If it is made by the authors, the script should be provided.

A: Thanks a lot. The references have added to the revised manuscript.

- According to the Figure S2A, the authors prepared 3 sgRNAs. However, mutation was observed at the specific locus on the genome (targeted by S3?). Did authors use a mixture of sgRNAs, or was one sgRNA used for each egg? This point should be explained clearly.

A: Thanks for your valuable suggestion! We are sorry that we did not describe it clearly. The mutation in the *Squid*^{-/-} genome was near the S2 target site, so it was mainly caused by S2. The three sgRNAs were thoroughly mixed with Cas9 protein and then injected into each silkworm eggs. We had revised the description in the revised manuscript.

Responses to reviewers' comments

Dear reviewers,

Thank you very much for your continued comments and professional advice. We appreciate the time and effort you have dedicated to reviewing our manuscript. We understand that some of their concerns remain, and we have carefully re-examined the areas you have highlighted. Your feedback is crucial to us, and we are committed to making the improvements to address your concerns fully.

Reviewers' comments:

Reviewer #2 (Remarks to the Author):

This is the revised version of a manuscript in which the role of Squid in silkworm midgut development was investigated. The paper has been largely revised and significantly improved. The authors added new experimental evidence to corroborate their results and revised the text detailing several unclear aspects and providing more information on the experimental approach. Thus, the overall quality of the product has been improved. However, I'm still concerned about the impairment of nutrient digestion and absorption after Squid knockout. I understood the authors' explanation on this issue but, although they provided qPCR results showing the reduced expression of some genes, these are mostly coding for factors involved in metabolism (glycolysis, pyruvate metabolism, etc.), not directly linked to nutrient digestion and absorption. Thus, I'm still wondering if there is a real effect on digestive and absorption properties of this organ after Squid knockout, and I think that this aspect needs to be carefully considered before publication.

A: We sincerely thank you for raising this important point regarding the direct impact of *Squid* knockout on nutrient digestion and absorption. Lipids accumulated in the egg during oogenesis represent the primary energy source for the developing embryo (Fruttero *et al.*, 2017). The embryonic midgut, which develops to encapsulate and subsequently digest these lipid reserves, is essential for nutrient and energy supply (Chapman, 2013). Our observations support this physiological timeline: on day 5, the midgut structure had begun to form and the midgut lumen appeared empty (Fig. R1Aa); by day 6, the midgut was fully developed and contained abundant lipid droplet-like vacuoles (Fig. R1Ab, yellow arrow); and by day 7, these lipid vacuoles were largely absent in the wild-type midgut lumen (Fig. R1Ac). This progression strongly suggests that lipids are actively digested and absorbed by the silkworm midgut during embryonic development. Critically, Nile Red staining revealed clear lipid droplet accumulation in the midgut lumen of *Squid*^{-/-} embryos at day 7, in contrast to wild-type controls (revised Fig. 3Aa–f), indicating impaired lipid clearance in the mutant. To assess whether lipid digestion and absorption were directly affected, we detected the expression of key lipid-processing genes. RT-qPCR showed that expression levels of *Lipase member H-A* were

significantly decreased in *Squid*^{-/-} embryos, whereas *Fabp* and *Apoltp* expression levels remained largely unchanged (Fig. R1B–D). Consistently, transcriptomic analysis indicated that downregulated DEGs in mutants were enriched in pathways related to nutrient absorption and energy supply (revised Fig. 3B). Although lipid transport protein expression was unaltered, the reduction in lipase expression is expected to directly impair lipid hydrolysis, thereby limiting the release of absorbable free fatty acids. Together, these results suggest that *Squid* deficiency severely disrupts midgut digestive function and nutrient absorption capacity. We have added the above results in the revised manuscript (Section Results; Fig. 3I–K and Supplementary Fig. S3A).

Figure R1. Midgut development during embryogenesis and expression of lipid-processing genes. (A) Morphological analysis of the embryonic midgut at indicated days post-oviposition in wild-type embryos. MG, midgut. L, lumen. Lipid droplet-like vacuoles are marked by orange arrow. Scale bars represent 500 μm or 100 μm, respectively. (B–D) Relative expression levels of genes (*Lipase member H-A*, *Fabp*, and *Apoltp*) involved in lipid digestion and absorption in wild-type and *Squid*^{-/-} embryos at day 7, quantified by RT-qPCR. Data are presented as mean ± SEM. *** $p < 0.001$.

References:

- Fruttero LL, Leyria J, Canavoso LE. Lipids in Insect Oocytes: From the Storage Pathways to Their Multiple Functions. *Results Probl Cell Differ*. 2017; 63:403-434.
- Chapman RF, The insects: structure and function. Simpson SJ, Douglas AE (eds) Cambridge University Press, Cambridge, 2013.

Reviewer #3 (Remarks to the Author):

In the revised manuscript, the authors provide new data to address my previous

comments. Some issues, such as insufficient methodological descriptions, have been clearly resolved.

However, revised manuscript does not address the critical problems I pointed out in the previous comment. Particularly, the evidence supporting the claim that Squid within the midgut cells contributes directly to midgut development remains weak. The authors have not ruled out the possibility that Squid indirectly regulates midgut development, as described below.

1) As I commented in my previous review, the authors have not resolved the issue that embryonic development in Squid knockout (KO) is delayed or arrested. Because of this, it remains possible that the abnormal midgut structure observed in Squid KO embryos is a consequence of entire developmental delay or arrest, rather than a midgut-specific developmental defect caused by the absence of Squid.

Although the authors state that midgut cell proliferation is completed by 7-day-old eggs in both wildtype (WT) and Squid KO embryos, based on phospho-HistoneH3 observation (Fig. S4, pp. 12-13, ll. 16-1), they do not provide the precise developmental timing at which midgut cell proliferation is completed in WT. Therefore, it remains possible that embryonic development in Squid KO is delayed or arrested after the midgut cell proliferation stage.

The authors should perform positive control assays for phospho-histoneH3 detection at earlier developmental stages (e.g. Day-4 to Day-6 embryos), rather than in larval wing disks. I also strongly recommend analyzing midgut development at earlier stages, at least in WT embryos, to demonstrate that the abnormal midgut structure in Squid KO (abnormal cell morphology and lipid accumulation) is distinct from that in early developmental stages in WT. Ideally, time-lapse analysis should be performed in Squid KO embryos, however, I understand that collecting homozygotic mutants prior to pigmentation may be challenging.

A: Thank you for this critical and insightful comment. Indeed, unlike *Drosophila*, which allows for tissue-specific genetic manipulations, it is challenging to conduct tissue-specific functional studies of genes in *Bombyx mori*, whether through CRISPR/Cas9-mediated knockout or RNA interference (RNAi). We fully agree that it is important to determine whether the midgut defects in *Squid* knockout embryos reflect a direct, tissue-autonomous role of Squid or are secondary to a general developmental delay. To address this issue, we performed additional phospho-histone H3 (P-H3) detection and HE staining at earlier wild-type embryonic midgut developmental stages (day 5-day 6), following the reviewer's suggestion. HE staining revealed that the wild-type midgut epithelium formed a single layer with an empty lumen on day 5 (Fig. R2Aa), developed into a well-defined pseudostratified structure with numerous undigested lipid droplet-like vacuoles in the lumen by day 6 (Fig. R2Ab), and exhibited a fully organized pseudostratified epithelium without obvious lipid droplet-like vacuoles by day 7 (Fig. R2Ac). In contrast, the *Squid*^{-/-} embryonic midgut epithelium appeared disorganized and contained abundant lipid-like vacuoles in the lumen at day 7 (Fig. R2Ad). Moreover, P-H3 detection showed strong nuclear signals in nearly all

midgut epithelial cells on day 5 (Fig. R2Ba-c, white arrow), sparse and scattered signals on day 6 (Fig. R2Bd-f), and complete absence by day 7 in wild-type embryos (Fig. R2Bg-i). Similarly, no P-H3 signal was detected in *Squid*^{-/-} embryos at day 7 (Fig. R2Bj-l).

Together, these data show that although the mutant midgut at day 7 accumulates lipids similarly to the wild-type day 6 midgut, it lacks the corresponding proliferation signal and fails to establish the organized pseudostratified epithelium seen in wild-type at day 6. These findings indicate that midgut development in *Squid* mutants is already impaired before day 6, rather than merely delayed. We have added the above results in the revised manuscript (Section Results; Fig. 2D-D' and Supplementary Fig. S3).

Figure R2. Analysis of midgut morphology and cell proliferation during

embryogenesis. (A) Representative semi-thin cross-sections of the wild-type embryonic midgut at days 5-7 and *Squid*^{-/-} embryonic midgut at days 7 post-oviposition, showing tissue architecture. MG, midgut. Orange arrows point to lipid droplet-like vacuoles within the lumen. The scale bar in a, b, c, and d represents 500 μm , 100 μm , 100 μm , and 100 μm , respectively. (B) Detection of cell proliferation signals in the midgut epithelium of wild-type from day 5 to day 7 and *Squid*^{-/-} embryos on day 7. Proliferating cells are indicated by white arrows. The yellow arrow represents the non-specifically stained peritrophic matrix. The scale bars represent 50 μm .

2) The role of Wnt signaling regulated by *Squid* and Axin in midgut development is unclear. In Figure 5E, β -catenin is localized to the cell membrane rather than the nucleus. β -catenin is known to localize to the membrane in the absence of Wnt signaling (Valenta et al., EMBO J., 2012). Thus, high abundance of β -catenin does not necessarily indicate that Wnt signaling is activated in WT embryo to induce cellular differentiation. The authors provide no evidence that 1) *Squid*-mediated increases in Axin-L enhance Wnt signaling, or 2) enhanced Wnt signaling contributes to midgut cellular differentiation. Furthermore, as noted by all reviewers, the authors did not provide any evidence that *Squid*-mediated increases in Axin-L occur within the midgut. Therefore, the relationship between *Squid* and Wnt signaling should be discussed more carefully. For example, this study does not “demonstrate” that *Squid* contributes to the Wnt/ β -catenin signaling pathway during embryonic midgut development (p. 19, ll. 22-23).

A: Thanks for your valuable comment.

1. We acknowledge the technical challenge of performing gene overexpression or tissue-specific manipulations in live silkworms, which currently limits our ability to conduct gain-of-function studies of *Squid* or to artificially enhance Wnt signaling in order to assess their roles in midgut cellular differentiation. In light of this, we have carefully revised our conclusions throughout the manuscript to clarify that the *Squid* mutation disrupts *Axin* alternative splicing, which in turn affects β -catenin levels specifically in the basal region of the midgut epithelium. These revisions have been incorporated in the Abstract, Introduction, Results, and Discussion sections.

2. As the reviewer rightly points out, several studies have shown that condensation of β -catenin can promote its clustering together with E-cadherin and α -catenin at the cell cortex, which facilitate the formation of nascent cell-cell junctions (Brembeck et al, 2004; Valenta et al, 2012; Monster et al., 2025). Thus, the β -catenin protein on the cell membrane also has the function in regulating cell structure, without relying on Wnt signaling. We have now included a discussion of this β -catenin-mediated structural role in the revised Discussion section.

3. The primary focus of this study is to establish the role of *Squid* in regulating *Axin* alternative splicing and the subsequent impact on β -catenin expression. Accordingly, we have refined the discussion section regarding *Squid* and the Wnt/ β -catenin pathway to more precisely reflect our findings. The specific mechanisms by which Wnt/ β -catenin signaling operates during embryonic midgut development remain an important open question and will be the subject of future investigation.

References:

- Brembeck FH, Schwarz-Romond T, Bakkers J, Wilhelm S, Hammerschmidt M, Birchmeier W. Essential role of BCL9-2 in the switch between beta-catenin's adhesive and transcriptional functions. *Genes Dev.*, 2004; 18(18):2225-30.
- Monster JL, Manzato C, van der Beek JA, Pannekoek WJ, Hummelink JA, Hadders MA, de Heus C, Klumperman J, Schuijers J, Gloerich M. β -catenin condensation facilitates clustering of the cadherin/catenin complex and formation of nascent cell-cell junctions. *Nat Commun.*, 2025; doi: 10.1038/s41467-025-66984-2. Epub ahead of print.
- Valenta T, Hausmann G, Basler K. The many faces and functions of β -catenin. *EMBO J.*, 2012; 31(12):2714-36.